# Dual Effect of Soloxolone Methyl on LPS-Induced Inflammation In Vitro and In Vivo

**DOI:** 10.3390/ijms21217876

**Published:** 2020-10-23

**Authors:** Andrey V. Markov, Aleksandra V. Sen’kova, Valeriya O. Babich, Kirill V. Odarenko, Vadim A. Talyshev, Oksana V. Salomatina, Nariman F. Salakhutdinov, Marina A. Zenkova, Evgeniya B. Logashenko

**Affiliations:** 1Institute of Chemical Biology and Fundamental Medicine, Siberian Branch of the Russian Academy of Sciences, 630090 Novosibirsk, Russia; andmrkv@gmail.com (A.V.M.); alsenko@mail.ru (A.V.S.); vlgrps@gmail.com (V.O.B.); k.odarenko@yandex.ru (K.V.O.); talyshev.v@gmail.com (V.A.T.); marzen@niboch.nsc.ru (M.A.Z.); 2N.N. Vorozhtsov Novosibirsk Institute of Organic Chemistry, Siberian Branch of the Russian Academy of Sciences, 630090 Novosibirsk, Russia; ana@nioch.nsc.ru (O.V.S.); anvar@nioch.nsc.ru (N.F.S.)

**Keywords:** 18βH-glycyrrhetinic acid, derivatives, soloxolone methyl, anti-inflammatory activity, cytokines production, LPS-induced endotoxemia, carrageenan-induced peritonitis, target prediction, molecular docking

## Abstract

Plant-extracted triterpenoids belong to a class of bioactive compounds with pleotropic functions, including antioxidant, anti-cancer, and anti-inflammatory effects. In this work, we investigated the anti-inflammatory and anti-oxidative activities of a semisynthetic derivative of 18βH-glycyrrhetinic acid (18βH-GA), soloxolone methyl (methyl 2-cyano-3,12-dioxo-18βH-olean-9(11),1(2)-dien-30-oate, or SM) in vitro on lipopolysaccharide (LPS)-stimulated RAW264.7 macrophages and in vivo in models of acute inflammation: LPS-induced endotoxemia and carrageenan-induced peritonitis. SM used at non-cytotoxic concentrations was found to attenuate the production of reactive oxygen species and nitric oxide (II) and increase the level of reduced glutathione production by LPS-stimulated RAW264.7 cells. Moreover, SM strongly suppressed the phagocytic and migration activity of activated macrophages. These effects were found to be associated with the stimulation of heme oxigenase-1 (HO-1) expression, as well as with the inhibition of nuclear factor-κB (NF-κB) and Akt phosphorylation. Surprisingly, it was found that SM significantly enhanced LPS-induced expression of the pro-inflammatory cytokines interleukin-6 (IL-6), tumour necrosis factor-α (TNF-α), and interleukin-1β (IL-1β) in RAW264.7 cells via activation of the c-Jun/Toll-like receptor 4 (TLR4) signaling axis. In vivo pre-exposure treatment with SM effectively inhibited the development of carrageenan-induced acute inflammation in the peritoneal cavity, but it did not improve LPS-induced inflammation in the endotoxemia model.

## 1. Introduction

Inflammation plays an important role in the biodefense system against harmful stimulation such as pathogens and injured/damaged tissue. This defense system consists of closely related innate and adaptive immune responses [1]. Among the various immune cells involved in this complex response, macrophages are unique because they are not only the most typical innate immune cells, they also trigger the adaptive immune response. Due to their immune surveillance role, macrophages sense a wide spectrum of stimuli, spanning from viral, microbial, and parasite antigens, immune complexes, and apoptotic or necrotic cells to various mediators released by other cells [2,3,4,5]. In response to these stimuli, macrophages are activated, which allows them to combat pathogens, exert an immunomodulatory role, and maintain tissue integrity [2,6,7].

Macrophages can be phenotypically polarized by surrounding microenvironmental stimuli and signals to mount specific functional programs [8]. Several forms of macrophages have been described in mice and humans based on the production of specific factors, expression of cell surface markers, and biological activities [8,9,10]. Polarized macrophages can be broadly classified into two main groups: classically activated macrophages (M1), which steer pro-inflammatory responses, and alternatively activated macrophages (M2), which drive immune regulation and tissue remodeling. M2 macrophages can be further subdivided in M2a, M2b, M2c, and M2d phenotypes based upon the applied stimuli and the resulting transcriptional changes [8,9,11,12,13].

In particular, macrophages are activated when exposed to inflammatory stimuli such as lipopolysaccharide (LPS). LPS, a pathogenic endotoxin present in the outer membrane of Gram-negative bacteria, promotes the inflammatory reaction and oxidative stress, and it is commonly used to generate models of disease for evaluation of the efficacy of anti-inflammatory drugs [14,15]. Macrophages exposed to LPS excessively produce pro-inflammatory mediators and cytokines as well as reactive oxygen species (ROS) [16,17,18]. Nitric oxide (II) (NO) and prostaglandin E2 (PGE2) are representative pro-inflammatory mediators. NO, regulated by inducible nitric oxide synthase (iNOS), reacts with peroxides to promote oxidative stress and inflammatory processes [19,20]. In addition, major pro-inflammatory cytokines, such as tumour necrosis factor-α (TNF-α), interleukin-6 (IL-6), and interleukin-1β (IL-1β), are overexpressed in macrophages stimulated by LPS and contribute to the pathogenesis of various inflammatory diseases [21,22]. Accumulating evidence suggests that LPS causes the overproduction of pro-inflammatory mediators and cytokines through the activation of nuclear factor-κB (NF-κB), which is associated with the phosphatidylinositol-3 kinase (PI3K)/Akt pathway [23,24,25]. In addition, inflammatory stress is known to increase reactive oxygen species (ROS) production and reduce the production of antioxidant enzymes that protect tissues from oxidative damage [26,27].

Currently, there are many in vivo models of LPS-induced inflammation, including endotoxemia [28,29], acute injury of the lungs [30,31], kidneys [32,33], and liver [34,35], neuroinflammation, and neurodegeneration [36,37]. The basis of all these models is a massive and sharp increase in the level of pro-inflammatory cytokines in the blood serum (septic shock) [38,39], which then causes damage to internal organs with multiple organ failure [40,41] and organ fibrosis as a result [37,42]. The severity of the manifestations of LPS-induced inflammation depends on the dose of LPS and the route of administration (intravenous, intraperitoneal, intramuscular, intranasal, etc.) [30]. The administration of high LPS doses leads to the death of the animal, whereas the administration of low doses leads to the development of multiple organ injury [43].

Despite the protective role of inflammation in eliminating pathogens and promoting tissue regeneration, the exacerbated inflammatory process is involved in several diseases in humans, including cardiovascular diseases, diabetes, arthritis, inflammatory bowel disease, and periodontitis, to mention only a few.

Natural products have been investigated as an alternative source of drugs to modulate the inflammatory process. Plants represent important sources of lead compounds, with up to 40% of modern drugs being based on plant materials [44]. Extracts from plants, which were later recognized as containing triterpenoids, have been used as medicinal treatments for ailments such as fever, infection, and inflammation since ancient times. Among them are semisynthetic derivatives of oleanolic acid CDDO (2-cyano-3,12-dioxooleana-1,9-dien-28-oic acid) and its structurally related analogues (CDDO-Me, CDDO-Im, CDDO-CN, etc.) [45,46,47]. All of these compounds have been reported to display various bioactivities in vitro and in vivo, including cytoprotection, cancer cell growth inhibition, apoptosis induction, and the inhibition of inflammation [45,46,47,48,49,50]. CDDO and CDDO-Me are currently in phase 3 clinical trials for chronic kidney diseases, Alport syndrome, and pulmonary hypertension (https://clinicaltrials.gov).

18βH-Glycyrrhetinic acid (18βH-GA) is a bioactive component abundant in liquorice root that possesses a great variety of pharmacological and biological properties such as antiviral, anti-ulcerogenic, anti-allergic, anti-oxidant, anti-hepatotoxic, anti-tumour, and anti-inflammatory activities [51,52]. Earlier, we reported that a semisynthetic derivative of 18βH-GA, soloxolone methyl (methyl-2-cyano-3,12-dioxo-18βH-olean-9(11),1(2)-dien-30-oate; SM) (Figure 1), obtained by direct modification of the A- and C-rings of 18βH-GA, exhibits high antiproliferative and pro-apoptotic activities in a number of cancer cell lines [53]. In addition, we showed that SM displays anti-inflammatory activity; SM caused a significant and dose-dependent decrease in NO production in LPS-activated J774 murine macrophages [54]. Moreover, we showed that SM inhibits the influenza virus A-activated expression of IL-6 and TNF-α and prevents the development of inflammatory changes in lung tissue caused by the virus in mice [55].

The aim of the present study was to explore the anti-inflammatory and anti-oxidative stress activities of SM in vitro and in vivo in the models of LPS-activated macrophages and LPS-induced endotoxemia and carrageenan-induced peritonitis, i.e., models of acute inflammation with different mechanisms. We showed in vitro that SM attenuates inflammation by inhibiting macrophage ROS and NO production, phagocytosis, and migration activity, which is accompanied by the downregulation of NF-κB and Akt activation; however, this was found to stimulate the secretion of pro-inflammatory cytokines. These data were confirmed in vivo using LPS-induced and carrageenan-induced inflammation models.

## 2. Results

### 2.1. Cytotoxicity of SM on RAW 264.7

To investigate the cytotoxic effect of soloxolone methyl (SM) alone and SM+LPS on RAW 264.7 murine macrophage cell line, the cells were incubated with SM at different concentrations in medium supplemented with LPS or not for 24 h followed by cell viability measurements using the MTT assay. The data displayed in Figure 2 demonstrate that SM at concentrations ≤ 1 µM both in the presence and in the absence of LPS had no effect on the viability of RAW 264.7 cells. It should be noted that LPS enhanced SM cytotoxicity: the IC_50_ values were 3.2 ± 0.2 and 1.7 ± 0.2 µM for SM used alone or with LPS, respectively. Therefore, SM concentrations ≤ 1 µM are not toxic to cells and were used in the subsequent experiments.

### 2.2. Effects of SM on Cellular Redox Imbalance and NO Production in RAW 264.7 Macrophages

Heme oxigenase-1 (HO-1) is the inducible isoform of hemeoxygenase and exerts significant anti-inflammatory and antioxidant effects. With respect to the anti-inflammatory activity of HO-1, it has been reported that an innate deficiency of HO-1 leads to severe inflammation, whereas HO-1 overexpression enhances anti-inflammatory effects [56,57,58]. Moreover, HO-1 expression plays a critical role in preventing inflammation in LPS-induced macrophages [59].

Recently, we showed that SM treatment significantly upregulates the expression of HO-1 in the human cervical carcinoma KB-3-1 cells [60]. Here, we studied whether SM could stimulate the production of HO-1 in RAW 264.7 macrophages. Cells were challenged with SM (0.25 and 0.5 μM) for 24 h, and the levels of HO-1 mRNA were quantified by RT-qPCR (Figure 3A). The data show that SM caused a significant upregulation of HO-1 mRNA expression (*p* < 0.005) in a dose-dependent manner: the HO-1 mRNA level was increased by 6- and 9-fold in cells treated with 0.25 and 0.5 μM SM, respectively (Figure 3A). Moreover, as expected, SM stimulated HO-1 protein expression to the same extent as it stimulated the expression of HO-1 coding mRNA (Figure 3 B).

LPS is known to induce ROS production in immune cells, which is a crucial event underlying LPS-elicited inflammatory responses. The excessive generation of ROS influences the antioxidant system, leading to a reduction in GSH levels [61]. The effect of SM on intracellular ROS formation in LPS-stimulated RAW 264.7 cells was analyzed using the 5,6-carboxy-2′7′-dichlorofluorescin diacetate (DCF-DA) assay (Figure 3C). As shown in Figure 3C, LPS alone evoked a significant (*p* < 0.01) increase in ROS generation by the cells, which was evident by the more than 2-fold increase in DCF fluorescence intensity. Cells treated with 0.5 μM SM suppressed LPS-induced ROS generation, but the level of ROS still remained 1.5-fold higher as compared to the control. Interestingly, SM was also able to slightly but significantly (*p* < 0.05) decrease ROS production in unstimulated macrophages (Figure 3C).

Since intracellular ROS readily interact with reduced GSH, which controls the intracellular redox status, the intracellular level of GSH was measured (Figure 3D). Assuming that the GSH content of the control macrophages was 100%, the GSH level was decreased to 59% ± 5% when the cells were stimulated with LPS. This reduced GSH level was restored by treatment with SM up to a value even higher than that in the control cells (120% ± 4%). The treatment of unstimulated cells with SM also led to a higher level of GSH (131% ± 9%).

Several studies have suggested that HO-1 signaling has a vital role in intracellular antioxidant systems and that crosstalk between HO-1 and iNOS regulates NO production [62,63]. iNOS is normally not expressed in cells, but macrophages exposed to bacterial endotoxin respond with increased expression of the iNOS enzyme, leading to NO generation [64]. Interactions between NO and ROS in cellular redox signaling are important determinants of the inflammatory response through effects on redox-sensitive gene expression. Therefore, to assess the effect of SM on the immune response, we analyzed iNOS expression and NO release by LPS-stimulated RAW264.7 macrophages incubated in the presence of SM.

iNOS mRNA expression, assessed by RT-qPCR after RAW 264.7 cells were exposed to LPS, SM and LPS+SM, is displayed in Figure 3E. SM used alone did not stimulate iNOS expression, and the level of iNOS mRNA was the same as in untreated control cells. As depicted in Figure 3E, iNOS was significantly (*p* < 0.001) upregulated under LPS treatment, but when LPS and SM were applied together, the level of stimulated iNOS expression was reduced and became close to that of untreated cells (the difference was statistically insignificant), clearly showing that SM blocks iNOS activation by LPS.

To address whether the inhibition of iNOS is linked to a reduction in NO release, NO production was measured in the form of nitrite in cell culture medium using the Griess reagent (Figure 3F). A dramatic increase in the concentration of secreted nitrite in cell culture medium was observed in LPS-stimulated macrophages, while SM significantly inhibited NO production by these cells in a dose-dependent manner (Figure 3F).

Taken together, these results show that SM prevents oxidative stress in LPS-stimulated RAW264.7 cells.

### 2.3. SM Inhibits the Phagocytic and Migratory Activity of Macrophages

Phagocytosis, a “cellular chewing” of foreign substances by macrophages, is an initial step of innate immunity. Phagocytosis assay was performed to evaluate the effect of SM on the phagocytic activity of macrophages. Phagocytic activity was analyzed by flow cytometry and then represented by mean fluorescence intensity (MFI) (Figure 4A). Data showed that LPS significantly (*p* < 0.01) increased the phagocytic function of RAW 264.7 cells as compared to control (MFI: 84507 for LPS vs. 59062 for the control). The addition of SM to LPS-induced RAW 264.7 cells reduced FITC–dextran uptake by 30% (MFI: 58843) and 50% (MFI: 40831) for SM 0.25 and 0.5 µM, respectively. The same effect was observed when SM was used with unstimulated cells (Figure 4A).

Previous studies have demonstrated that LPS mediates inflammatory responses through increasing cell migration to sites of inflammation [65]. To investigate the possible effects of SM on LPS-stimulated RAW 264.7 motility, cells were allowed to migrate in the CIM plate for 48 h in the presence of one or both stimuli in the upper chamber (Figure 4B). The level of cell migration was calculated as the migration index (MI) = cell index (control)/cell index (experiment). The data show that LPS used alone increased MI by 1.5-fold. The treatment of LPS-stimulated cells with 0.25 µM SM reduced macrophage motility to some extent, while SM at 0.5 µM reduced cell migration to a level even lower than that of untreated cells (MI 0.8 vs. 1 for LPS+SM 0.5 µM and untreated control, respectively, *p* < 0.05).

### 2.4. Effect of SM on Pro-Inflammatory Cytokines in LPS, Poly(I:C), and IFN-γ-Stimulated RAW 264.7 Cells

Cytokines have a variety of functions in the immune system, including the recruitment and activation of immune cells. It has been reported that pro-inflammatory cytokines secreted by LPS-stimulated macrophages include IL-1β, IL-6, and TNF-α [66], which indicate the occurrence of inflammation.

The mRNA expression levels of TNF-α, IL-1β, and IL-6 were assessed by RT-qPCR analysis in LPS-stimulated cells treated with SM (Figure 5A–C). The mRNA levels of each of these pro-inflammatory cytokines were upregulated by LPS. Moreover, 2- to 5-fold higher levels of mRNA of these cytokines compared with LPS only were observed when LPS-stimulated cells were treated with SM; the highest induction of expression by SM was observed for IL-6 mRNA (Figure 5A).

To determine whether this induction of inflammatory signaling corresponded to stimulated pro-inflammatory cytokine secretion, we evaluated by ELISA the secretion of TNF-α and IL-6 in activated RAW264.7 macrophages after treatment with SM. As shown in Figure 5D–E, resting RAW264.7 macrophages secrete IL-6 and TNF-α at a very low level. The treatment of these cells with SM or LPS alone enhanced IL-6 secretion, and SM increased this level at least twice over the level of LPS. Cells treatment with both SM and LPS significantly (*p* < 0.01) enhanced the secretion of IL-6 and TNF-α in a dose-dependent manner in comparison with controls; moreover, the production of IL-6 was stimulated to a greater extent as compared to TNF-α. Thus, our data show that SM treatment enhances pro-inflammatory cytokine production in LPS-stimulated macrophages.

To determine if the increase in IL-6 production by LPS-stimulated cells observed after treatment with SM is linked with the nature of inflammatory stimuli, RAW 264.7 cells were also activated with poly (I:C) or IFN-γ and incubated with SM. As shown in Figure 4F,G, co-treatment of cells with a combination of poly (I:C) or IFNγ and SM also led to an increase in IL-6 expression, but to a lesser extent as compared to the combination with LPS. The increase in the mRNA level was 9-fold for SM+LPS vs. 3-fold for SM+poly (I:C) (Figure 5A,F), and at the protein level, measured by ELISA, the increase was 7-fold for SM+LPS vs. 2-fold for SM+IFNγ (Figure 5E,G).

### 2.5. Effect of SM on the Activation of the Akt, NF-κB, and AP-1 Signaling Pathways and TLR4 in LPS-Induced RAW264.7 Cells

Since exposure to LPS activates the Akt signaling pathway in macrophages [67,68], we tested the effect of SM on the LPS-induced phosphorylation of Akt in RAW264.7 cells by Western blot analysis. As shown in (Figure 6A,B), LPS significantly activated Akt in macrophages, while incubation with SM did not change the phosphorylation of this kinase, leaving it at the level observed in unstimulated cells. However, SM inhibited by 1.5-fold the phosphorylation of Akt in LPS-stimulated cells.

NF-κB is a transcription factor that regulates the expression of inflammatory mediators. To investigate the effect of SM on NF-κB signaling, we determined the nuclear translocation of NF-κB. In resting cells, NF-κB is located in the cytoplasm as an inactive complex bound to its inhibitor, inhibitor of kappa B-α (IκB-α). Once induced by LPS, IκB-α is rapidly phosphorylated by IκB-α kinase and degraded, which results in the translocation of NF-κB into the nucleus. As shown in Figure 6A,C cells treated with SM alone did not change the IκB-α level. The degradation of IκB-α was observed in RAW264.7 cells stimulated with LPS, while the incubation of LPS-stimulated cells with SM inhibited this degradation to some extent but did not completely abolish it (Figure 6A,C). However, analysis of IκB-α phosphorylation revealed a clear inhibitory effect of SM on LPS-induced IκB-α phosphorylation (Figure 6A,D). Since p65 is a major component of the NF-κB complex, we investigated whether SM prevents the translocation of the p65 subunit of NF-κB from the cytosol to the nucleus after its release from complex with IκB-α (Figure 6E,F). LPS caused a decrease in cytosolic p65 and an increase in nuclear p65 (Figure 6E,F); treatment with SM attenuated levels of p65 in the nuclear fraction and increased the cytoplasmic level of NF-κB p65 to the level of resting untreated cells.

These data show that SM acts as a negative regulator of LPS-stimulated NF-κB and Akt activation in RAW 264.7 macrophages. Considering that NF-kB is a master regulator of NO production by macrophages, we can conclude that the ability of SM to inhibit the LPS-induced expression of iNOS and NO production by inflamed macrophages is associated with suppression of the NF-κB signaling pathway.

Along with NF-κB, the transcription factor AP-1 is known to be involved in the transcriptional regulation of inflammatory responses [69,70,71,72]. Therefore, we investigated the effect of SM on AP-1 (c-Jun) mRNA expression by RT-qPCR. As shown in Figure 6G, treatment of RAW264.7 cells with LPS alone increased the level of c-Jun mRNA by 1.5-fold, and the incubation of macrophages with SM+LPS increased this level by 2.5-fold. So, we demonstrated the ability of SM to positively regulate the transcription factor AP-1 in RAW264.7 cells.

LPS initiates the inflammatory cascade in macrophages via Toll-like receptor 4 (TLR4). The stimulation of TLR4 activates multiple signaling pathways via the phosphorylation of Akt, c-Jun NH2-terminal kinases (JNK), extracellular signal regulated kinase (ERK1/2), and p38 mitogen-activated protein kinases (MAPKs). In turn, these signaling pathways mediate multiple downstream events, leading to the activation of AP-1 and NF-κB, which coordinate the induction of a range of inflammatory proteins. Therefore, we decided to analyze the effect of SM on the expression of TLR4. RT-qPCR analysis revealed that SM significantly increased the expression of this receptor in LPS-challenged macrophages, whereas LPS without SM did not affect TLR4 levels (Figure 6H).

Our findings reveal that the stimulatory effect of SM on LPS-induced pro-inflammatory cytokine production in macrophages is regulated by the c-Jun/TLR4 signaling axis.

### 2.6. Effects of SM on Macrophage Surface Marker Expression

We showed that SM exhibits anti-oxidative activity in LPS-stimulated macrophages by inhibiting ROS and NO production, via the induction of HO-1 and deactivation of the NFκB pathway, as well as by reducing phagocytosis and migration. However, at the same time, treatment with SM, especially together with LPS, led to a significant increase in the secretion of the pro-inflammatory cytokines IL-6 and TNF-α.

Macrophages under LPS treatment can differentiate to the M1 or M2b (if LPS is used with the immune complex) phenotype. LPS triggers a number of transcriptional events in macrophages that can be seen in surface molecular markers such as the co-stimulatory proteins CD80 and CD86 [3], the LPS adaptor protein CD14, and mannose receptor 1 CD206. We decided to measure the expression levels of these molecules using flow cytometry (Figure 7).

The data show that LPS stimulates the expression of CD80, CD86, and CD14 (Figure 7A–C), but not CD206 (Figure 7D) on RAW264.7 macrophages. The treatment of macrophages with SM leads to an increase of expression of CD80, CD86, CD14, as well as CD206 and SM activates the expression of mentioned macrophage surface markers to a greater extent in comparison with LPS. These results suggested that CD80 and CD86 are involved in the immune regulation of RAW264.7 cells by SM. So, we can hypothesize that under SM action, macrophages could shift differentiation to an M2b-like phenotype, but other experiments to support this idea are needed.

### 2.7. Anti-Inflammatory Effect of SM In Vivo

Earlier, we showed that SM exhibited anti-inflammatory activity in animal models with primary local inflammation, such as influenza virus A-induced pneumonia [55], as well as carrageenan and histamine-induced paw oedema [73]. Here, the anti-inflammatory effect of SM was studied in vivo using models of LPS-induced endotoxemia and carrageenan-induced peritonitis, both of which are models of acute inflammation, but with different triggers and mechanisms. LPS-induced endotoxemia is a model of general inflammation, where the main mechanism is the cytokine storm, i.e., a massive and rapid increase in pro-inflammatory cytokines in the blood serum [74,75]. Carrageenan-induced peritonitis is a model with initial local inflammation in the peritoneal cavity followed by the development of a general inflammatory response [76,77]. However, despite the differences in the initiation of these inflammatory models, both LPS-induced endotoxemia and carrageenan-induced peritonitis proceed via TLR4 activation [78,79].

#### 2.7.1. LPS-Induced Endotoxemia

We investigated the anti-inflammatory effect of SM in non-lethal and lethal endotoxemia models. In non-lethal endotoxemia, LPS was administered at a dose sufficient to cause the general inflammatory response, but not the animal’s death. In lethal endotoxemia, LPS was administered at a dose causing the gradual death of animals. The dosages of LPS for these models were chosen according to published data [80,81,82].

Dexamethasone, a known anti-inflammatory drug, was chosen as the reference substance, because the chemical structure of 18βH-GA is similar to that of hormones, and it has been shown that 18βH-GA can bind glucocorticoid receptors and mineralocorticoid receptors, although the binding affinities are much lower than those between glucocorticoid receptors and dexamethasone [83].

Figure 8A shows that an intraperitoneal administration of LPS at a dose of 5 mg/kg (non-lethal endotoxemia) significantly enhanced the secretion of TNF-α in the serum of experimental mice in comparison with healthy animals. SM administrated at a dose 50 mg/kg to mice with LPS-induced non-lethal endotoxemia decreased the TNF-α level by 2.1-fold, but the differences were statistically insignificant. A similar decrease in the TNF-α level in LPS-challenged mice was observed for the dexamethasone, so no difference in the TNF-α level between dexamethasone and SM was found (Figure 8A). It is to be noted that the administration of SM to healthy mice did not influence TNF-α secretion. The important aspect of assessing non-lethal endotoxemia was that the administration of SM caused an increase in the serum level of IL-6. It should be emphasized that in this model of LPS-induced inflammation, the increase in the IL-6 level is robust (Figure 8B, LPS + vehicle), and SM administration to LPS-challenged mice additionally showed a trend toward increased IL-6 expression (*p* = 0.04), while dexamethasone administration slightly but significantly (*p* < 0.005) reduced the IL-6 level in the serum of experimental animals (Figure 8B).

Thus, in contrast to the strong stimulation by SM secretion of pro-inflammatory cytokines by macrophages in vitro (see above), the administration of SM to healthy unchallenged mice did not increase the levels of pro-inflammatory cytokines in the serum (Figure 8A,B) and did not significantly alter the level of these cytokines in LPS-challenged mice.

The effect of SM on LPS-induced mortality was assessed by measuring the survival rate of mice challenged with 20 mg/kg of LPS (lethal endotoxemia). As shown in Figure 8C, SM exhibited no protective effect in lethal endotoxemia in comparison with dexamethasone. The follow-up of SM-treated and untreated LPS-challenged mice was similar (Figure 8C). Some insignificant effect on survival rates observed in the group of vehicle-treated mice could be attributed to the non-specific activation of immune response by the vehicle itself.

#### 2.7.2. Carrageenan-Induced Peritonitis

Carrageenan-induced peritonitis is a model of acute inflammation based on increased vascular permeability and leukocyte migration into the peritoneal cavity caused by a phlogogen (carrageenan) [76]. Mice were intraperitoneally administered with SM in 10% Tween-80 (vehicle) at a dose of 50 mg/kg 1 h prior to peritonitis induction by 1% carrageenan followed by the counting of leukocytes in peritoneal fluid 4 h after carrageenan injections. Dexamethasone (0.5 mg/kg) was used as a reference drug.

The results depicted in Figure 9A,B show a 6-fold increase in the total leukocyte number in the peritoneal exudate 4 h after carrageenan injection in vehicle-treated group compared with the healthy group injected with saline buffer. Peritoneal exudates were predominately composed of neutrophils (Figure 9A,B). The treatment of carrageenan-challenged animals with SM or dexamethasone significantly decreased the migration of inflammatory cells to the peritoneal cavity compared with the control vehicle-treated group (Figure 9A,B). SM administrated prior to carrageenan injection promoted a significant inhibition of the inflammatory process, with a 7-fold decrease in the number of leukocytes in peritoneal exudates, mainly due to a reduction in the number of neutrophils (Figure 9A,B, left panel). The exudates in this case were represented predominately by lymphocytes, similar to the healthy group (Figure 9A,B, right panel).

The treatment of mice with dexamethasone also decreased leukocyte migration with a 5.8-fold reduction in the total leukocyte count. However, dexamethasone did not normalize the cell ratio, and the peritoneal exudates in this group were represented mainly by neutrophils (Figure 9A,B, right panel). Thus, the data clearly show that pre-exposure administration of SM effectively inhibited the development of carrageenan-induced acute inflammation.

However, despite the clear anti-inflammatory effect of SM on the total and differential leukocyte count of peritoneal exudates in carrageenan-induced peritonitis, the test compound did not affect the level of pro-inflammatory cytokines in peritoneal exudates in comparison with carrageenan-challenged mice without treatment (Figure 9C), while the reference compound dexamethasone reduced TNF-α and IL-6 to the level of healthy animals (Figure 9C). So, this finding agrees well with the effects of SM on the production of IL-6 and TNF-α observed in the LPS-induced endotoxemia model.

## 3. Discussion

Inflammation is a central event of the host defense against pathogenic mediators such as damaged cells or invading microbes, which plays a fundamental role in immune modulation and the restoration of tissue homeostasis [84]. During inflammation, an uncontrolled overproduction of pro-inflammatory substances can contribute to a variety of pathophysiological conditions and metabolic disorders [85]. Given the complicated mechanisms triggering and controlling inflammation and the presence of unwanted side effects followed by therapy with modern anti-inflammatory drugs [86], the search for and development of novel anti-inflammatory compounds, especially those that display multitarget effects on different inflammation-related signaling pathways, are important tasks of medicinal chemistry.

Macrophages play a central role in all stages of the inflammatory response [87] and, moreover, they are involved in the development of a wide range of chronic inflammatory-associated diseases, e.g., diabetic nephropathy in type 2 diabetes [88] or cartilage degradation in osteoarthritis [89]. Thus, the analysis of efficiency of novel anti-inflammatory candidates in macrophages is a necessary step of their preclinical evaluation. According to published data, LPS-challenged murine RAW264.7 macrophages are one of the most widely used models for this aim [90,91]. LPS promotes inflammation by inducing the production of reactive oxygen species, an effect balanced by antioxidant enzymes such as HO-1 [92,93], which subsequently modulates innate [94] and adaptive [95] immune cell function. Oxidative stress caused by LPS triggers the activation of macrophages, leading to an excessive inflammatory process [96]. LPS induces ROS production, which is a crucial event underlying LPS-elicited inflammatory responses. The excessive generation of ROS influences the antioxidant system, leading to a reduction in GSH levels. The link between LPS-induced inflammation and oxidation-reduction reactions [97,98,99] predicts that agents with combined anti-inflammatory and antioxidant properties may be advantageous in attenuating or even preventing endotoxemia. The important sources of such compounds are natural metabolites and their semisynthetic derivatives; additionally to pronounced inhibitory effects on oxidative and inflammatory stresses, these molecules display relatively low systemic toxicity in comparison to chemically synthesized agents [100].

Previously, we showed that a semisynthetic derivative of 18βH-glycyrrhetinic acid (18βH-GA), soloxolone methyl (SM), bearing a cyano-enone pharmacophore in ring A (Figure 1), significantly decreased the production of NO in LPS-challenged J774 macrophages in vitro [101] and displayed marked anti-inflammatory activity in murine models of acute carrageenan-induced paw oedema [73] and influenza A-stimulated pneumonia [55]. This work aimed to extend our knowledge about the effects of SM on the macrophage component of inflammation and to reveal the mechanisms underlying its modulatory activity in the LPS-induced inflammatory response in vitro and in vivo.

Given that the release of ROS and reactive nitrogen species by macrophages in response to pathogens is rapid and massive and their uncontrolled production at inflamed sites causes serious damage to tissues [102], we analyzed the effects of SM on oxidative stress-associated parameters in LPS-activated RAW264.7 macrophages. Our findings clearly demonstrate that SM displayed marked antioxidant potential, as the treatment of macrophages by SM significantly decreased the generation of ROS in both unstimulated and LPS-stimulated cells (Figure 3C) and the production of NO by inflamed macrophages (Figure 3F). In order to more accurately interpret the obtained results, we investigated the effects of a semisynthetic triterpenoid on the expression of heme oxigenase-1 (HO-1), which is a key antioxidant enzyme that, according to our recent report, was significantly susceptible to SM in another cellular model [60], and inducible nitric oxide synthase (iNOS). We showed that the treatment of LPS-challenged RAW264.7 cells with SM caused a significant up-regulation of HO-1 (Figure 3A,B) and reduced iNOS expression (Figure 3E), which is consistent with the revealed ability of SM to inhibit the release of ROS and NO by macrophages. In addition, SM restored the GSH level, which had been depleted in LPS-stimulated cells, to a level comparable with that of intact control cells (Figure 3D). These results agree well with published data demonstrated that cyano-enone-bearing triterpenoids (CETs), including SM’s position isomer CDDO-Me, CDDO-Im, NZ, dh404, and SO1989, effectively inhibit the release of NO and iNOS expression in LPS/IFNγ-activated RAW264.7 cells [103,104,105,106,107,108,109] and LPS-challenged primary peritoneal macrophages [110], decreased the production of ROS, and activated the expression of HO-1 in unstimulated [105,107], LPS-stimulated, and tert-butyl hydroperoxide-stimulated RAW264.7 cells [106,107,108,109] as well as LPS-induced neutrophils [111,112]. Moreover, CDDO-Me and CDDO-MA were found to increase the production of GSH in human primary astrocytes [113] and block 3-NP neurotoxin-induced striatal GSH depletion in rats, respectively [114].

According to published data, the NF-κB pathway is a key CET-sensitive signaling pathway underlying the anti-inflammatory activity of these compounds in immune cells [103,106,108,109]. Additionally, PI3K/Akt signaling, playing a critical role in the regulation of the inflammatory response in TLR-stimulated macrophages [115], was also reported to be susceptible to CDDO-Me in RAW264.7 cells [103]. The analysis showed that SM, similar to its analogues, effectively suppressed the NF-κB pathway in LPS-challenged RAW264.7 macrophages by inhibiting the phosphorylation and subsequent degradation of IκB (Figure 6C,D) and nuclear translocation of the p65 subunit of NF-κB (Figure 6F); it also blocked the activation of Akt kinase (Figure 6A,B). Considering that NF-κB is a well-known master regulator of iNOS expression [116], we conclude that the ability of SM to inhibit NF-κB signaling determines its inhibitory effects on the iNOS/NO system in endotoxin-activated macrophages. The observed up-regulation of HO-1 and increased production of GSH in SM-treated RAW264.7 cells (Figure 3A,B) can be explained by the activation of Nrf2 pathway. Previously, it was shown that CETs are powerful inducers of the transcription factor Nrf2 and Nrf2-dependent activation of the endogenous antioxidant machinery [105,106,107,111,112,117]. It was shown that the anti-inflammatory and antioxidative actions of an entire set of synthetic triterpenoids, including CDDO, are closely linked, and that the Nrf2/ARE system seems to provide a common mechanism for these activities of the triterpenoids. Thus, our findings suggest that the inhibitory effect of SM on LPS-induced oxidative stress in macrophages can be mediated by multitarget mechanisms.

It is known that LPS stimulates not only the massive release of reactive species by macrophages but also significantly enhances the phagocytic and chemotactic activities of these cells [118]. Our results clearly show that SM effectively inhibited these processes in both unstimulated and LPS-stimulated RAW264.7 macrophages (Figure 4), indicating the ability of SM to block different macrophage functions. It should be emphasized that the suppressive activity of SM on macrophages phagocytosis and motility described in this work was revealed for the first time for this class of compounds. CDDO-Me was shown to display only a minimal effect on the phagocytic activity of LPS-activated microglial BV-2 cells [110], and no effect on this process was observed in LPS/IFNγ-treated RAW264.7 cells [109]. The ability of CETs to modulate macrophage migration, as far as we know, has not yet been published. Only Kim et al. [119] showed the inhibitory activity of CDDO-Me on monocyte infiltration in the inflamed frontoparietal cortex in mice; however, this effect was more associated with the CDDO-Me-induced alteration in the chemokine profile than its influence on the motility of cells.

Despite the known master regulatory role of NF-κB in controlling macrophage phagocytosis [120] and revealed inhibitory effects of SM on NF-κB signaling in inflamed macrophages (Figure 6C–F), we yet inferred that the observed SM-induced blockade of phagocytic activity in RAW264.7 cells is NF-κB-independent. As depicted in Figure 4A and Figure 6, the treatment of macrophages with SM without LPS significantly inhibited the uptake of FITC–dextran by the cells in comparison with the control; however, it did not alter the phosphorylation of IκB and nuclear translocation of the p65 NF-κB subunit. Moreover, Yang et al. [109] showed recently that SM’s position isomer CDDO-Me markedly inhibited NF-κB signaling in LPS/IFNγ-challenged RAW264.7 cells, but it does not modulate the phagocytosis in this model. Revealed disagreements between our data and previously reported results, describing the effects of CETs on phagocytosis, can be explained by different concentrations of compounds used in these studies. In our work, SM was used at 0.5 µM, whereas the effects of CDDO-Me in BV-2 microglial cells and RAW264.7 macrophages was investigated at markedly lower concentrations (0.1 µM) [109,110]. As shown by To el al. [121], another SM analogue, CDDO-Im at a high concentration (1 µM), can disorganize the microtubule network and dysregulate branched actin polymerization in Rat2 fibroblasts by directly binding to actin-related protein 3 (Arp3) [122]. Considering the important role of Arp3 in the regulation of both phagocytosis [123] and cellular motility [122] and the immediate participation of the microtubule cytoskeleton in these processes [124,125], we speculate that the observed inhibitory activity of SM on the phagocytic and migratory activities of RAW264.7 cells can be explained by its probable effects on these targets.

In order to analyze the effects of SM on macrophage functions more comprehensively, we evaluated its influence on the production of pro-inflammatory cytokines in LPS-stimulated RAW264.7 cells. Surprisingly, our findings showed that SM significantly enhanced the LPS-induced expression of IL-6 and TNF-α at both the mRNA and protein levels, as well as IL-1β mRNA expression in this model (Figure 5A–E). Moreover, we found that SM alone increased the production of IL-6 in resting macrophages but to a significantly lesser extent than LPS (Figure 5D). These results are mostly inconsistent with published data because it has been shown that CETs significantly inhibit LPS-induced expression of pro-inflammatory cytokines not only in macrophages [103,104,105,106,110] but also in microglia [110], peripheral blood mononuclear cells [112], and neutrophils [111]. In a recent study [126], CDDO-Me was shown to markedly enhance the LPS-stimulated production of TNF-α in macrophages; however, in contrast to our work, where macrophages were classically activated (M1 phenotype), these findings were obtained in M2-polarized cells. In order to understand whether the observed SM-stimulated induction of cytokines is TLR4-dependent, we analyzed the effects of SM on the expression of IL-6 in RAW264.7 macrophages, activated by poly (I:C) and IFN-γ, the ligands of TLR3 and interferon-γ receptor, respectively. The data clearly show that the cytokine-induced activity of SM is independent of TLR4, as SM reinforced the synthesis of IL-6 in both poly I:C- and IFN-γ-stimulated cells (Figure 5F,G), which is similar to LPS-challenged macrophages. However, the observed synergistic effects of SM and poly (I:C)/IFN-γ in RAW264.7 cells was markedly less intensive that that of SM and LPS (compare Figure 5F/A and Figure 5G/D).

As noted above, two key signaling cascades were identified previously as targets of CETs, which is associated with their inhibitory effects on LPS-induced inflammation, including the NF-κB [103,106] and Nrf2 [105,106,111,112] pathways, which are inhibited and activated by these compounds, respectively. NF-κB plays a critical role in multiple aspects of innate and adaptive immune functions, including the induction of various pro-inflammatory cytokines [127]. The transcription factor Nrf2, a key regulator of the antioxidant response, can also directly control inflammatory processes [128]. Considering that SM effectively inhibited NF-κB (Figure 6C–F) and probably activated Nrf2, i.e., an induction of Nrf2-dependent HO-1 and GSH (Figure 3A,B,D) [129,130] in LPS-challenged RAW264.7 cells, the central question raised by these data is which mechanisms underlie the ability of SM to enhance the synthesis of pro-inflammatory cytokines by macrophages in response to LPS.

In order to understand whether SM increases the sensitivity of RAW264.7 cells to LPS, we analyzed its effect on the expression of TLR4. RT-qPCR analysis revealed that SM significantly increased the expression of this receptor in LPS-challenged macrophages, whereas LPS without SM did not affect the TLR4 level (Figure 6H). These findings can partly explain the synergistic effects of SM and LPS on the induction of cytokines in RAW264.7 cells. However, the question remained as to which signaling pathway can mediate both the observed up-regulation of TLR4 in SM-treated macrophages (Figure 6H) and signal transduction from activated TLR4 to the genes encoding pro-inflammatory cytokines, taking into account that NF-κB signaling was blocked by SM (Figure 6C–F). In our previous report, we showed that AP-1 is one of the key master regulators controlling the response of KB-3-1 cervical carcinoma cells to SM [60]. Given this fact, the involvement of AP-1 in the regulation of TLR4 expression and pro-inflammatory cytokines [131] and the regulatory role of c-Jun, a key component of AP-1, in modulating the effects of a structural analogue of SM (RTA408) on the IL-1β-induced inflammatory response in rat astrocytes [132], we analyzed the effect of SM on the expression of c-Jun in LPS-treated RAW264.7 cells. The analysis showed that SM significantly induced c-Jun expression in macrophages in comparison to LPS-challenged control cells (Figure 6G), demonstrating the ability of SM to positively regulate AP-1 in RAW264.7 macrophages. These results are consistent with published data on the up-regulation of c-Jun and c-Fos detected in KB-3-1 cells and HUVEC treated with SM and CDDO-Im, respectively [60,133]. Thus, our findings demonstrate that the stimulatory effect of SM on LPS-induced pro-inflammatory cytokine production in macrophages is regulated by the c-Jun/TLR4 signaling axis. However, in order to elucidate this mechanism more precisely, additional analysis of the effects of SM on the protein levels of c-Jun and TLR4 is required.

Another explanation for the observed elevation in the level of pro-inflammatory cytokines under SM treatment might arise from the fact that SM may influence macrophage polarization. Macrophage polarization between the M1 and M2 phenotypes is an important mechanism for the regulation of inflammatory responses. These macrophages differ in their cell surface markers, secreted cytokines, and biological functions. However, studies have indicated that the induction routes and regulated biological processes are a complex interlacing network rather than a simplistic scheme [134]. We showed that SM treatment of RAW264.7 cells enhances the expression of the surface markers CD80, CD86, and CD206 (Figure 7) compared to LPS exposure. CD206, also termed MRC1 (C-type mannose receptor 1), is an M2 macrophage marker in both the mouse and the human [135,136]. So, we can suppose that SM has a tendency to polarize macrophages to the M2 phenotype. On the other hand, as mentioned above, we observed an increase in the level of pro-inflammatory cytokines under SM treatment (Figure 5A–E). It is known that M2 macrophages can be further subdivided in M2a, M2b, M2c, and M2d based on the applied stimuli and the resultant transcriptional changes [8,9,11,12,13]. The M2b subset of macrophages is an exception, and it retains high levels of inflammatory cytokine production such as IL-6, IL-1β, and TNF-α [8,137]. Based on the expression profile of cytokines, chemokines, and other secreted factors, M2b macrophages regulate the breadth and depth of the immune response and the inflammatory reaction [137]. Despite various important roles in many diseases, the definite and specific molecular markers of M2b macrophages have not yet been unified and established. It has been shown that CD86, which works in tandem with CD80, is expressed by M2b macrophages, is considered a marker for this subtype [138], and is a suitable marker for discriminating M2b from the other subtypes of M2 macrophages [139,140]. To summarize, based on the enhanced expression of CD206 together with CD86/CD80 and the increased level of pro-inflammatory cytokines, we can hypothesize that SM polarizes macrophages to the M2b phenotype, but additional experiments to confirm or deny this hypothesis are needed.

It is unlikely that macrophages exist in clear distinct phenotypes, especially in vivo. Instead, there probably coexist intermediate activated subsets, because many stimuli are present at the same time in tissues. Interestingly, the polarization of macrophages may depend on cellular types and stimuli, and the same stimuli can promote pro- or anti-inflammatory macrophages under different circumstances.

In order to shed some light on the possible molecular mechanisms of observed immunostimulatory effects of SM in macrophages and to generate novel ideas for further studies in this field, we attempted to reveal hypothetical primary protein targets of SM using in silico approaches. Firstly, in order to identify proteins, the activation or inhibition of which can stimulate LPS-induced inflammatory response, text mining analysis was performed, and the obtained hit proteins are listed in Table 1.

CD36 was added to this list because Ikeda et al. [147] showed previously that ursolic acid induces the release of pro-inflammatory IL-1β from murine peritoneal macrophages as a result of its direct interaction with CD36. Next, to evaluate the ability of SM to bind to the listed proteins, molecular docking simulations were carried out. The results (Figure 10A,B) clearly show that OGT and MerTK can be considered molecular targets of SM: triterpenoid can dock well into the active sites of mentioned proteins, which is characterized by binding energies comparable with that of known OGT and MerTK inhibitors (Figure 10A) and forming hydrogen bonds with key residues needed for the activity of these enzymes (Figure 10B) [148,149,150]. The analysis of the docked pose of SM in CD36 showed that triterpenoid can interact with the receptor in the phosphatidylserine recognition region; the present results show that SM formed a hydrogen bond with Asp270 (3.21 Å) and a non-bonding interaction with Asp118, which is crucial for binding of phosphatidylserine to CD36 [151]. Given that phosphatidylserine is a well-known “eat-me” signal exposed on the surfaces of apoptotic cells [152] and that apoptotic cells inhibit the LPS-mediated secretion of pro-inflammatory mediators by macrophages [153], we infer that the binding of SM to CD36 can decrease the efficacy of recognition of apoptotic cells by macrophages that can contribute to the observed reinforcement of LPS-induced inflammatory response in SM-treated RAW264.7 cells compared to LPS-activated untreated macrophages. As depicted in Figure 10A, SIRT1 and EP4 cannot be considered as primary protein targets of SM, because SM binding with these proteins is characterized by a positive ΔG value in the case of SIRT1 and ΔG = −6.3 kcal/mol for EP4, which is higher than our threshold of −7.0 kcal/mol.

Furthermore, in order to estimate the interconnection between SM-sensitive pro-inflammatory genes/proteins and revealed hypothetical primary targets of SM (OGT, MerTK and CD36), a protein–protein interaction (PPI) network was reconstructed from (a) IL-6 and TNF-α, being up-regulated in LPS-treated macrophages in response to SM at both the mRNA and protein levels, (b) the c-Jun/TLR4 signaling axis, (c) OGT, MerTK, and CD36, and (d) the Top-100 protein partners of the mentioned proteins in the STRING database (Figure 10C). It was found that the listed proteins formed a tight regulome, which is characterized by 3217 interconnections between nodes (Figure 10C). Analysis of the reconstructed PPI network showed that other SM-sensitive genes/proteins, identified by us in RAW264.7 cells, including Rela (p65 NF-κB), IL-1β, Akt, IκB, CD14, CD80, and CD86, were also involved in the regulome (Figure 10C, orange color), which indicates the reliability of the probable mechanism of action of SM determined using the in silico approach.

Further analysis of the number of protein partners between hypothetical protein targets of SM and c-Jun and downstream TLR4/IL-6/TNF-α showed that their involvement in the regulation of pro-inflammatory response decreased in the order: c-Jun>CD36>MerTK>OGT (Figure 10D). According to the presence of edges between OGT/MerTK/CD36 and c-Jun (Figure 10D), revealed probable targets of SM can be characterized by an upstream position against the transcription factor AP-1. Thus, the performed bioinformatic analysis clearly showed the expediency of further investigation of OGT, MerTK, and CD36 as probable master regulators, controlling the immunostimulatory potential of cyano-enone-bearing triterpenoids. However, it should be noted that the obtained results are based on computer simulation approaches and give only a novel hypothesis about the molecular mechanism underlying the synergistic action of SM with LPS on the production of pro-inflammatory cytokines in macrophages, and the validation of this hypothesis is a task of our further studies.

The ability of SM to significantly enhance the IL-6-stimulating effect of LPS, revealed in RAW264.7 macrophages in vitro (Figure 5A,D), was also demonstrated by us in non-lethal LPS-induced endotoxemia in vivo (Figure 8B). Moreover, it was found that a single i.p. injection of SM at the non-toxic dosage of 50 mg/kg increased the mortality of LPS-challenged mice compared to vehicle-treated animals (Figure 8C). Although the observed decline in the survival rate was statistically insignificant (log-rank *p* = 0.14), our findings in both lethal and non-lethal LPS-induced endotoxemia models agreed well with our data obtained on LPS-challenged macrophages. Interestingly, SM, similarly to dexamethasone, decreased the level of TNF-α in the serum of LPS-injected mice; however, the effect induced by both compounds was statistically insignificant (Figure 8A). Considering the results and the significant difference between the survival rates of SM- and dexamethasone-treated LPS-challenged mice (log rank *p* = 0.0004) (Figure 8C), we infer that the observed inhibition of TNF-α production by these compounds was not critical to their effects on the progression of endotoxemia. As in the case of SM’s effects on the expression of pro-inflammatory cytokines in macrophages, our results obtained in LPS-induced endotoxemia models are mostly inconsistent with published data. Previously, it was shown that CDDO-Me and CDDO-Im effectively reduced mortality in mice challenged by a lethal dosage of LPS [111,154]. A comparison of CET doses used in these studies confirmed our hypothesis regarding the dosage-dependent peculiarities of the effects of CETs on the LPS-induced inflammatory response: CDDO-Me and CDDO-Im increased the survival rate of LPS-challenged mice at a dose of ≈2.0 [111] and 1.6 mg/kg [154], respectively, whereas SM was used at a significantly higher dose (50 mg/kg). Thus, our findings in both cellular and animal models demonstrate that the usage of high non-toxic concentrations of CETs can lead not to the suppression of LPS-stimulated cytokine production, but, on the contrary, to its reinforcement. However, it is necessary to test this hypothesis more thoroughly.

In order to analyze the effects of SM on another independent model of TLR4-mediated inflammatory response in vivo, we studied its impact on carrageenan-induced peritonitis, the development of which consists of several phases, including not only an increase in the levels of pro-inflammatory cytokines, but also the activation of histamine/kallikrein/kinin and prostaglandin/COX-2 systems and the generation of neutrophil-derived reactive oxygen species, nitric oxide, and peroxynitrite [155,156,157]. Our results show that SM (50 mg/kg), in contrast to LPS-induced inflammation, effectively suppressed inflammatory process in this model by inhibiting the migration of leukocytes to the inflamed peritoneal cavity (Figure 9A,B) and, moreover, restoring the leukocyte subtype content in the peritoneum, similar to non-inflamed control group (Figure 9C). However, despite the clear anti-inflammatory effect of SM on the leukocyte count in carrageenan-induced peritonitis, this compound did not affect the level of pro-inflammatory cytokines in peritoneal exudates in comparison with the untreated carrageenan-challenged mice (Figure 9C), which agreed well with the effects of SM on the production of IL-6 and TNF-α, as revealed in the LPS-induced endotoxemia model (Figure 8A,B).

We consider that the revealed differences in the modulatory effects of SM on LPS- (Figure 8) and carrageenan-stimulated (Figure 9) inflammatory responses can be explained by the peculiarities in the mechanisms of the models. Considering that the requirement of leukocytes (mainly, neutrophils) in the inflamed peritoneal cavity with a subsequent massive release of ROS and RNS and oxidative stress-provoked tissue damage is the main mechanism of carrageenan-induced acute peritonitis [77], we infer that the ability of SM to effectively trigger the anti-oxidative response (Figure 3) and inhibit the motility of inflammatory cells (Figure 4B) can underlie its anti-inflammatory effect on peritonitis, whereas its efficacy in endotoxemia, which is characterized by hyperinflammation, general septic status [158], and a tight association between severity and induction of the cytokine storm [159], was completely lacking.

It should be noted that hyperexpression of pro-inflammatory cytokines, especially IL-6, is not always detrimental to the organism. In a range of studies, it was found that IL-6 displays a marked protective role in a wide spectrum of inflammatory-associated pulmonary disorders, including pneumonia induced by influenza A [160,161], Mycobacterium tuberculosis [162], and Streptococcus pneumonia [163], as well as endotoxin- and ischemia-induced acute respiratory distress syndrome [164], LPS-induced acute lung injury [165], and lung tissue remodeling and fibrosis [160,166]. Thus, our data show that SM treatment may be advantageous in the control of local inflammation, especially lung inflammatory diseases.

## 4. Materials and Methods

### 4.1. Cell Culture and Semisynthetic Triterpenoids SM

The murine macrophage/monocyte cell line RAW 264.7 was kindly provided by Professor D. Kuprash (Engelhardt Institute of Molecular Biology, Moscow, Russia). Cells were maintained as monolayers in Dulbecco’s modified Eagle’s medium (DMEM) (GIBCO BRL, Gaithersburg, MD, USA) supplemented with 4.5 mg/mL glucose containing 10% foetal bovine serum, 60 U/mL penicillin, and 100 μg/mL streptomycin at 37 °C in 5% CO_2_.

The chemical synthesis of the 18βH-glycyrrhetinic acid derivative soloxolone methyl (SM) has been described before [53]. This compound has been completely characterized by chemical analysis and nuclear magnetic resonance. SM was dissolved in DMSO (10 mM), and stock solutions were stored at −20 °C.

### 4.2. Assessment of Cell Viability

To evaluate the cytotoxicity of SM, RAW 264.7 cells were seeded in 96-well plates at a density of 1 × 10^3^ cells/well. Cells were incubated under standard conditions for 24 h followed by their treatment with various concentrations of SM with or without *E. coli* LPS (1 µg/mL, serotype 055:B5, Sigma-Aldrich Inc., St. Louis, MO, USA) for 24 h. After the incubation was complete, MTT (Sigma-Aldrich Inc., St. Louis, MO, USA) was added to each well at 0.5 mg/mL, followed by incubation at 37 °C in the dark. After 2 h of incubation, the MTT solution was removed and 100 μl DMSO was added to dissolve the crystals. The absorbance of each well was read at test and reference wavelengths of 570 and 620 nm, respectively, on a Multiscan RC plate reader (Labsystems, Vantaa, Finland). The optical density of the formazan formed in the control (untreated) cells was considered to represent 100% viability.

### 4.3. Measurement of Cell Migration Using xCelligence Technology

RAW 264.7 cells were seeded in tetraplicate at 2 × 10^4^ cells per well in the upper chamber of CIM plates with the presence or absence of SM at 0.25 or 0.5 µM with or without *E. coli* LPS (1 µg/mL, serotype 055:B5, Sigma-Aldrich Inc., St. Louis, MO, USA). To stimulate cell migration, the lower chamber of the CIM plate contained 10% FBS as the chemoattractant. Cell index (electrical impedance) was monitored by the RTCA DP xCelligence System (ACEA Biosciences, Inc., San Diego, CA, USA) every 1 h for 48 h.

### 4.4. Phagocytic Uptake

RAW 264.7 cells were treated with 0.25 or 0.5 μM SM with or without LPS (1 µg/mL). After 18 h, the cells were incubated in culture medium with FITC–dextran (1 mg/mL) for 1 h at 37 °C or at 4 °C for the negative control. The reaction was stopped by the addition of ice-cold phosphate-buffered solution (PBS). After fixing the cells with 3.7% formaldehyde, phagocytic uptake was analyzed using an ACEA NovoCyte^TM^ flow cytometer (ACEA Biosciences Inc., San Diego, CA, USA). The uptake of FITC–dextran was expressed as Δ mean fluorescence intensity (MFI), i.e., MFI (uptake at 37 °C)—MFI (uptake at 4 °C).

### 4.5. Measurement of NO Production

RAW 264.7 cells were treated with 0.5 µM SM with or without *E. coli* LPS (1 µg/mL, serotype 055:B5, Sigma-Aldrich Inc., USA) for 24 h. Controls were maintained under the same culture conditions; however, they were not treated or stimulated. NO levels were indirectly determined by measuring the stable NO catabolite nitrite in the medium utilizing the Griess reaction. In brief, the conditioned medium (100 μl) was mixed with the same volume of Griess reagent (Promega, Madison, WI, USA) and incubated for 10 min at room temperature. The optical density at 540 nm was measured using a Multiscan RC plate reader (Labsystems, Vantaa, Finland) reader, and the nitrite concentration was calculated according to a standard curve generated from known concentrations of sodium nitrite.

### 4.6. ELISA for Pro-Inflammatory Cytokines

The generation of pro-inflammatory cytokines TNF-α and IL-6 was measured using ELISA kits. The RAW 264.7 cells were treated with various concentrations of SM with or without LPS (1 µg/mL) for 24 h, and cytokine contents in the cell-free supernatants were measured using murine TNF-α or IL-6 ELISA kits (Thermo Scientific, Rockford, IL, USA) according to the manufacturer’s protocols.

### 4.7. Real-Time Quantitative PCR (RT-qPCR) Assay for mRNA Levels

The mRNA expression of inflammatory factors in each group were detected by RT-qPCR. Total RNA was isolated from macrophages with Trizol reagent (Ambion, Carlsbad, CA, USA) according to the manufacturer’s instructions. cDNAs were synthesized by using a first-strand synthesis kit (RevertAid^TM^, Fermentas, Canada) for 1 h at 42 °C. Then, aliquots of the cDNAs were amplified with specific primers for inflammatory cytokines, respectively. Glyceraldehyde 3-phosphate dehydrogenase (GAPDH) primers were used as an internal control. The primer sequences of TNF-α, IL-6, IL-1β, HO-1, iNOS, TLR4, c-Jun, and GAPDH were as follows:iNOS forward, 5′-AAGGTCTACGTTCAGGACATC-3′;iNOS reverse, 5′-AGAAATAGTCTTCCACCTGCT-3′;HO-1 forward, 5′-ACAGATGGCGTCACTTCGT-3′;HO-1 reverse, 5′-GTGAGGACCCACTGGAGGA-3′;IL-6 forward, 5′-CCGGAGAGGAGACTTCACAG-3′;IL-6 reverse, 5′-TCCACGATTTCCCAGAGAAC-3′;TNF-α forward, 5′-TCAGCCTCTTCTCATTCCTG-3′;TNF-α reverse, 5′-TGAAGAGAACCTGGGAGTAG-3′;IL-1β forward, 5′-TGCAGAGTTCCCCAACTGGTACATC -3′;IL-1β reverse, 5′-GTGCTGCCTAATGTCCCCTTGAATC -3′;cJUN forward, 5′-ACGACCTTCTACGACGATGC-3′;cJUN reverse, 5′-CCAGGTTCAAGGTCATGCTC-3′;TLR4 forward, 5′-AGATCTGAGCTTCAACCC-3′;TLR4 reverse, 5′-AGTCCAGAGAAACTTCCTG-3′;GAPDH forward, 5′-ACCCCCAATGTGTCCGTCGT-3′;GAPDH reverse, 5′-TACTCCTTGGAGGCCATGTA-3′

The RT-qPCR conditions were 94 °C for 5 min followed by 40 cycles (95 °C for 30 s, 59 °C for 30 s, and 72 °C for 30 s). The relative expression of genes was normalised to GAPDH.

### 4.8. Protein Isolation and Western Blot Analysis

RAW 264.7 cells were incubated with SM at the indicated concentrations with or without LPS (1 µg/mL). The cells were collected, lysed with whole-cell lysis buffer (45 mM HEPES pH 7.5, 400 mM NaCl, 1 mM EDTA, 10% glycerol, 0.5% NP-40). Protein concentration was determined using a Bradford protein assay kit (Bio-Rad Laboratories, Hercules, CA, USA). In a parallel experiment, cytoplasmic and nuclear extracts were prepared using an NE-PER nuclear and cytoplasmic extraction reagent kit (Pierce; Thermo Fisher Scientific, Inc., Rockford, IL, USA) following the manufacturer’s instructions. For Western blotting, equal amounts of protein samples (30 μg/lane) were subjected to 10–12% sodium dodecyl sulphate-polyacrylamide gel electrophoresis (SDS-PAGE) and then transferred onto PVDF membranes (MP Biomedicals, Santa Ana, CA, USA). Subsequently, the membranes were blocked with 5% non-fat dry milk in Tris-buffered saline containing 0.1% Triton X-100 (TBST) for 1 h and probed with specific primary antibodies against HO-1, Akt, p-Akt, NF-κB, IκBα, p-IκBα, β-actin, and lamin B at 4 °C overnight. After washing with TBST, the membranes were incubated with the appropriate horseradish peroxidase (HRP)-conjugated secondary antibodies for 1 h at room temperature. After successive washes, the membranes were developed using a chemiluminescence reagent kit (Abcam, Eugene, OR, USA) according to the manufacturer’s instructions.

### 4.9. Determination of Intracellular ROS

Intracellular ROS production was monitored using 5,6-carboxy-2′7′-dichlorofluorescin diacetate (DCF-DA). This substrate freely permeates cells, and upon incorporation, it is oxidized to fluorescent DCF. RAW 264.7 cells were treated with 0.5 μM SM with or without LPS (1 µg/mL). After 24 h of incubation, the cells were stained with 10 μM DCF-DA (Molecular Probes, Eugene, OR, USA) for 30 min at 37 °C in the dark. The cells were collected, washed with PBS twice, and a total of 10,000 events were then immediately analyzed using an ACEA NovoCyte^TM^ flow cytometer (ACEA Biosciences Inc., San Diego, CA, USA).

### 4.10. Glutathione Estimation

RAW 264.7 cells were treated with 0.5 μM SM with or without LPS (1 µg/mL). After 24 h of incubation, the GSH level was quantified using the GSH-Glo glutathione assay (Promega, Madison, WI, USA) according to the manufacturer’s instructions.

### 4.11. Evaluation of CD14, CD206, CD80, and CD86 Expression

RAW 264.7 cells were treated with 0.5 μM SM or with LPS (1 µg/mL). After 24 h of incubation, cells were collected, the cell surfaces were blocked with 15% sheep serum at 4 °C for 15 min and then washed twice with phosphate-buffered solution (PBS, pH 7.2). Cells were further incubated with fluorescence-conjugated monoclonal antibodies, anti-mouse CD80 conjugated with allophycocyanin (CD80-APC), anti-mouse CD86 conjugated with Pacific Blue (CD86-PB), or anti-mouse CD206 conjugated with fluorescein isothiocyanate (CD206-FITC) for 30 min at room temperature. Following incubation with specific antibodies, cells were washed twice with PBS and resuspended in PBS. The MFI was determined for 10,000 cells in each sample using an ACEA NovoCyte^TM^ flow cytometer (ACEA Biosciences Inc., San Diego, CA, USA).

### 4.12. Molecular Docking

The docking of SM with SIRT1 (Protein Data Bank (PDB) ID: 4I5I), OGT (PDB ID: 6MA3), MerTK (PDB ID: 5U6C), EP4 (PDB ID: 5YWY), and CD36 (PDB ID: 5LGD) was performed using Autodock Vina [167]. The 3D structures of the mentioned proteins were uploaded from the RCSB Protein Data Bank (https://www.rcsb.org/), followed by the extraction of co-crystallized ligands from the uploaded PDB files, the addition of the polar hydrogens, and Gasteiger charges into the protein structures using AutoDock Tools 1.5.7. The 3D structure of SM and optimization of its geometry with the MMFF94 force field was carried out using Marvin Sketch 5.12 and Avogadro 1.2.0, respectively. All rotatable bonds within the SM’s structure were allowed to rotate freely. The used docking parameters are listed in Table 2.

The best molecular interactions were further identified based on the binding orientation of the proteins’ key residues and their binding energy values. The results were imported and analyzed using Discovery Studio Visualizer v. 19.1.0.18287 and LigPlot+ v. 1.4.5 (3D and 2D plot reconstruction, respectively).

### 4.13. PPI Network Reconstruction

The protein–protein interaction (PPI) network was reconstructed based on data deposited in the STRING database, containing protein pairs collected from five different sources, including high-throughput lab experiments, genomic context prediction, co-expression, text mining, and known PPIs from other databases [168]. SM-sensitive IL-6, TNF-α, TLR4, and c-Jun revealed probable SM primary targets as OGT, MerTK, and CD36, which were used as input nodes. A confidence score ≥ 0.7 and maximum additional interactors = 100 were set as the cut-offs. The reconstructed PPI network was visualized using Cytoscape v. 3.7.2. The numbers of the shortest paths (length ≤ 1 node) between probable targets of SM and SM-sensitive genes were identified and visualized using the Pesca v. 3.0 plugin [169] and Circos (http://mkweb.bcgsc.ca/tableviewer/) [170], respectively.

### 4.14. Mice

Outbred ICR female mice (30–35 g) were provided by the vivarium of the Institute of Chemical Biology and Fundamental Medicine SB RAS. Mice were kept at 5 mice per cage in a natural light regime with free access to food and water. Experiments were carried out in accordance with the European Communities Council Directive 86/609/CEE. The experimental protocols were approved by the Committee on the Ethics of Animal Experiments at the Institute of Cytology and Genetics SB RAS (protocol #51 from 23 May 2019).

### 4.15. LPS-Induced Endotoxemia

ICR mice were intraperitoneally (i.p.) administered with SM in 10% Tween-80 at the dose of 50 mg/kg and vehicle (10% Tween-80) 1 h prior to endotoxemia induction by LPS (055:B5, Sigma-Aldrich, USA) i.p. at a dose of 5 mg/kg (non-lethal endotoxemia) or 20 mg/kg (lethal endotoxemia). We used SM at a dose of 50 mg/kg, since in our previous work [73] for the evaluation of anti-inflammatory activity of SM on the models of carrageenan- and histamine-induced paw edema, we found out that the dose of 50 mg/kg is the maximum possible dose for SM that can be administered in mice without significant toxicity. Blood samples were collected from the retro-orbital sinus 4 h after LPS injection at a dose of 5 mg/kg. Blood serum was prepared by a standard protocol, and the pro-inflammatory cytokines TNF-α and IL-6 were measured by a mouse TNF-α ELISA kit and a mouse IL-6 ELISA kit (Thermo Scientific, Rockford, IL, USA) according the manufacturer’s instructions. The survival rates were evaluated after LPS injection at the dose of 20 mg/kg. The mortality of mice was assessed every 12 h for 7 days. SM- and vehicle-treated mice without LPS challenge were used as controls. In both experiments, dexamethasone at the dose of 0.5 mg/kg (non-lethal endotoxemia) and 1 mg/kg (lethal endotoxemia) was used as a reference drug with proven anti-inflammatory activity.

### 4.16. Carrageenan-Induced Peritonitis

ICR mice were intraperitoneally (i.p.) pretreated with SM in 10% Tween-80 (50 mg/kg), vehicle (10% Tween-80), or dexamethasone (0.5 mg/kg) as a reference anti-inflammatory drug 1 h prior to peritonitis induction by 1% carrageenan i.p. Saline buffer was also administered i.p. in the normal control group instead of carrageenan injection. Four hours after peritonitis induction, mice were sacrificed by cervical dislocation, and the peritoneal cavity was washed with 2 mL of heparinized cold saline buffer to obtain peritoneal exudates. The collected samples were centrifuged (2500 rpm, 5 min, 4 °C), the cell pellets were resuspended in 50 μl of PBS, and total leukocyte counts were performed with a Neubauer chamber by optical microscopy after diluting in Turk solution (1:20). The supernatants were collected to assess the levels of pro-inflammatory cytokines TNF-α and IL-6 using a mouse TNF-α ELISA kit and a mouse IL-6 ELISA kit (Thermo Scientific, Rockford, IL, USA) according to the manufacturer’s instructions. To determine the differential leukocyte counts, peritoneal cells were centrifuged and placed onto slides, stained with azur-eosin by the Romanovsky–Giemsa method, and examined by optical microscopy. The results were expressed as the number of total leukocytes (×10^6^/mL) and the ratio of neutrophils, lymphocytes, and monocytes (%).

### 4.17. Statistical Analysis

The data are expressed as the mean  ±  SD. The statistical analysis was performed using the two-tailed unpaired t-test. *p*-values less than 0.05 were considered statistically significant. Survival was estimated using the Kaplan–Meier method, and comparisons between groups were done using the log-rank test.

## 5. Conclusions

In summary, we found that a cyano-enone-containing triterpenoid SM displayed a dual effect on LPS-induced inflammatory response in RAW264.7 cells. On the one hand, SM effectively blocked endotoxin-triggered oxidative stress in macrophages, reducing ROS and NO production and restoring the GSH content to the control level, in addition to inhibiting the phagocytic and chemotactic activities of these cells, which was accompanied by suppression of the NF-κB signaling pathway and Akt activation. However, on the other hand, this triterpenoid significantly enhanced the LPS-induced expression of pro-inflammatory cytokines IL-6, TNF-α, and IL-1β in RAW264.7 cells via activation of the c-Jun/TLR4 signaling axis. The analysis of the animal models independently confirmed our data obtained in macrophages. SM was found to significantly inhibit carrageenan-induced peritonitis, where the infiltration of inflammatory cells into the peritoneal cavity and oxidative stress (two processes, sensitive to SM in RAW264.7 cells) play a key role in the progression of peritoneal inflammation, whereas SM had no protective effect on LPS-induced endotoxemia, the severity of which mainly depends on cytokine storm induction. Moreover, the latter was slightly exacerbated with SM treatment. Taken together, our results provide new insights into the understanding of the immunomodulatory effects of CETs and showed for the first time that these compounds at high non-toxic concentrations can display synergistic effects with LPS in the induction of cytokine expression in M1-polarized macrophages. Our findings confirm the strong capability of SM to treat local inflammation and the necessity for a comprehensive analysis of the septic status of patients before their involvement in the clinical investigation of CETs.

## Figures and Tables

**Figure 1 ijms-21-07876-f001:**
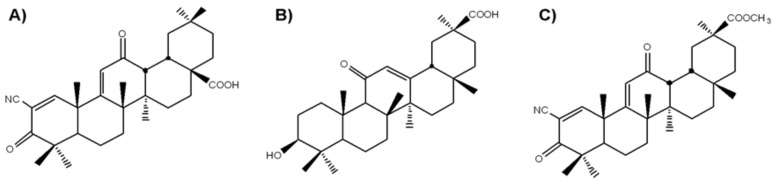
Chemical structure of CDDO (2-cyano-3,12-dioxooleana-1,9-dien-28-oic acid) (**A**); 18βH-GA (18βH-glycyrrhetinic acid) (**B**); soloxolone methyl (methyl-2-cyano-3,12-dioxo-18βH-olean-9(11),1(2)-dien-30- oate; SM) (**C**).

**Figure 2 ijms-21-07876-f002:**
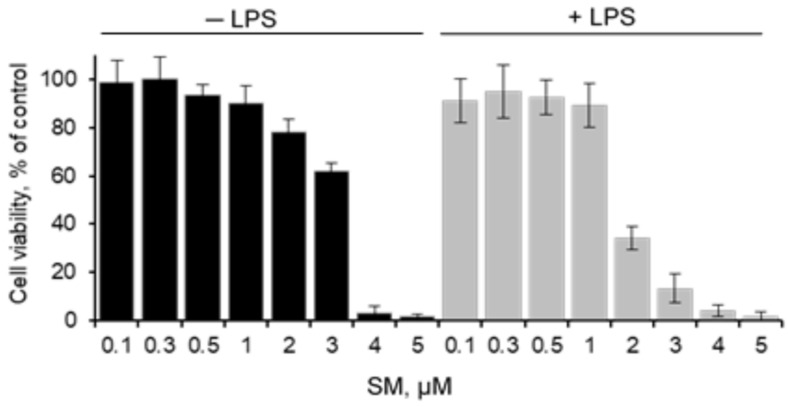
Effect of SM on the viability of RAW 264.7 macrophages. Cells were incubated with various concentrations of SM in medium supplemented or not with lipopolysaccharide (LPS) (1 µg/mL) for 24 h. Cell viability was assessed using the MTT assay, and the results are expressed as the percentage of viable cells in comparison with the viability of control (untreated cells). Values are expressed as the mean ± SD of three independent experiments.

**Figure 3 ijms-21-07876-f003:**
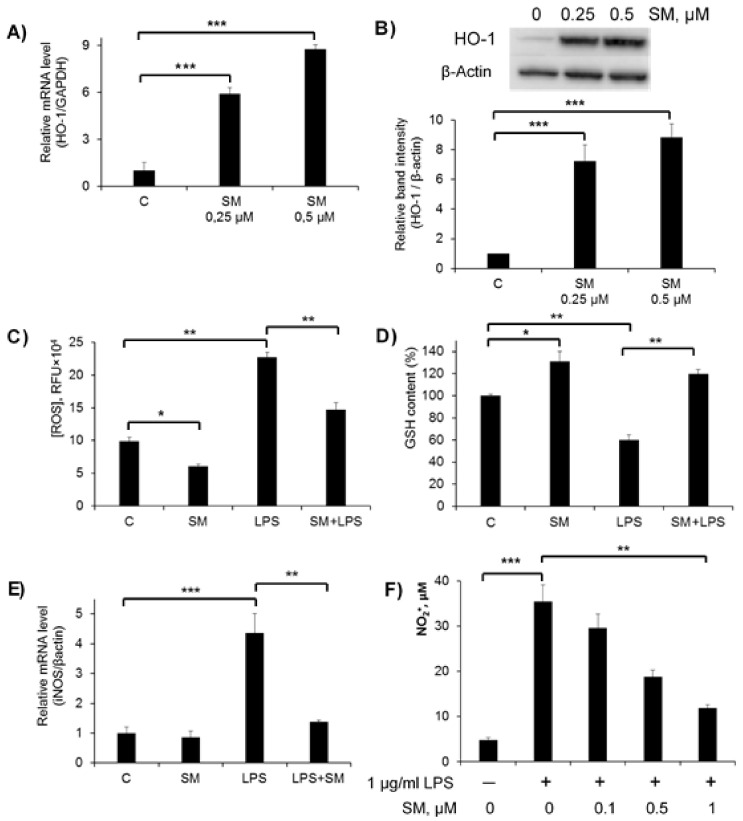
Anti-oxidant and anti-nitric oxide (NO) effects of SM in RAW 264.7 macrophages. RAW264.7 cells were incubated with or without the indicated concentrations of SM for 24 h. After incubation, (**A**) Heme oxigenase-1 (HO-1) mRNA levels were analysed by real-time PCR and (**B**) HO-1 protein levels were determined by Western blotting (upper panel, representative Western blot bands. Lower panel, summarized bar graph shows band intensity presented as the ratio of HO-1 over β-actin.). Cells were incubated with SM (0.5 µM), LPS (1 µg/ml), or SM+LPS for 24 h. After incubation, reactive oxygen species (ROS) (**C**) and GSH (**D**) production was measured using the 5,6-carboxy-2′7′-dichlorofluorescin diacetate (DCF-DA) assay or the GSH–Glo glutathione assay, respectively; the inducible nitric oxide synthase (iNOS) mRNA level was evaluated by RT-qPCR at an SM concentration of 0.5 µM (**E**), and the NO concentration in the cell culture supernatant was measured via the Griess reaction (**F**). In all experiments, untreated cells were used as the control (designated by C). Values are expressed as the mean ± SD of three independent experiments; * *p* < 0.05, ** *p* < 0.01, *** *p* < 0.005.

**Figure 4 ijms-21-07876-f004:**
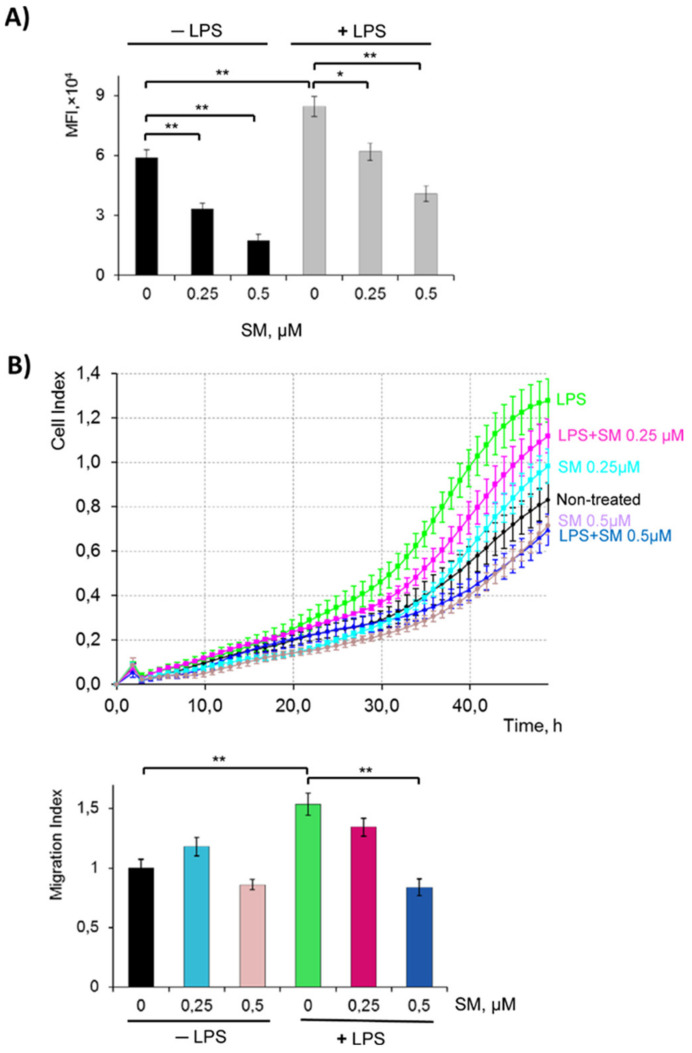
Effect of SM on phagocytosis (**A**) and migration dynamics (**B**) of RAW 264.7 cells. (**A**) Phagocytic ability was estimated using FITC–dextran (1 mg/mL). Uptake was assessed by flow cytometry. (**B**) For migration, RAW 264.7 cells were seeded in CIM plates and incubated with SM (0.5 or 0.25 µM), LPS (1 µg/mL) or SM+LPS, and migration was monitored continuously from the upper to lower chamber, indicated by the cell index (CI). Upper panel, representative trace from the RTCA-DP software showing macrophage migration toward the lower chamber; lower panel, Migration Index value after 48 h of incubation calculated as described in the Section 4. * *p* < 0.05, ** *p* < 0.01.

**Figure 5 ijms-21-07876-f005:**
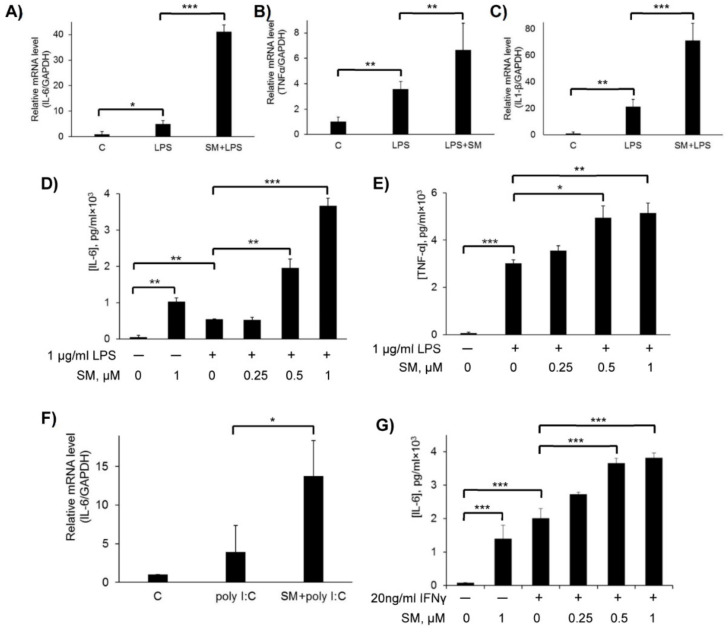
Effects of SM on the expression of inflammatory mediators. Raw 264.7 cells were treated with SM (0.5 µM), LPS (1 µg/mL), or SM+LPS (**A**–**E**) poly (I:C) and SM+poly I:C (**F**) or IFNγ and SM+ IFNγ (**G**) for 24 h. After incubation, mRNA levels of interleukin-6 (IL-6) (**A**,**F**), tumour necrosis factor-α (TNF-α) (**B**), and interleukin-1β (IL1-β) (**C**) were analyzed by RT-q PCR. The secretion of IL-6 (**D**,**G**) and TNF-α (**E**) in cell culture supernatants was analyzed by ELISA. In all the experiments, untreated cells were used as the control (designated by (**C**)). Values are expressed as the mean ± SD of three independent experiments; * *p* < 0.05, ** *p* < 0.01, and *** *p* < 0.001.

**Figure 6 ijms-21-07876-f006:**
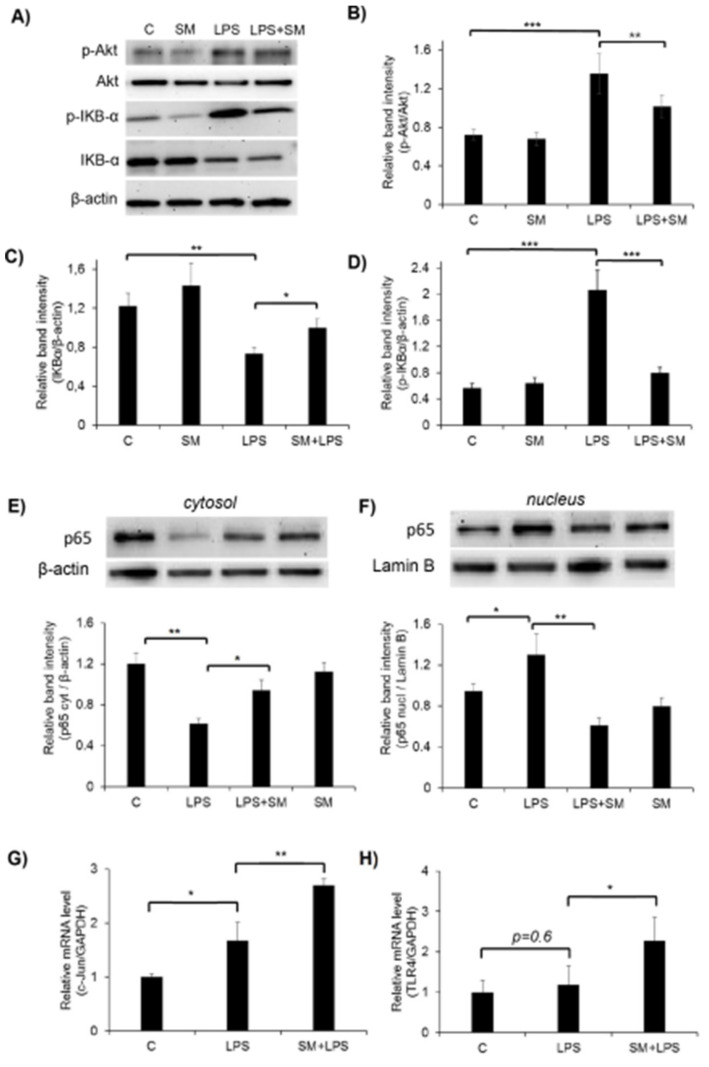
Effects of SM on the activation of Akt, nuclear factor-κB (NF-κB), AP-1 signaling pathway, and TLR4 in LPS-induced RAW264.7 cells. RAW264.7 cells were incubated with SM (0.5 µM), LPS (1 µg/mL), or SM+ LPS for 6 h. Phosphorylation and total protein expression of Akt and IKBα were analyzed in whole cell lysates using Western blot. (**A**) Representative Western blot bands; the summarized bar graph shows band intensity presented as the ratio of pAkt to Akt (**B**), inhibitor of kappa B-α (IκB-α) to β-actin (**C**), and pIκBα to β-actin (**D**). Cytoplasmic (**E**) and nuclear (**F**) fractions were prepared to analyze the nuclear translocation of p65. β-Actin was used as the loading control for the whole lysate and cytoplasmic fractions and lamin B1 were used for the nuclear fraction. c-Jun (**G**) and Toll-like receptor 4 (TLR4) (**H**) mRNA levels were analyzed by RT-qPCR. The data represent three independent experiments. In all experiments, untreated cells were used as the control (designated by C). * *p* < 0.05, ** *p* < 0.01, *** *p* < 0.001.

**Figure 7 ijms-21-07876-f007:**
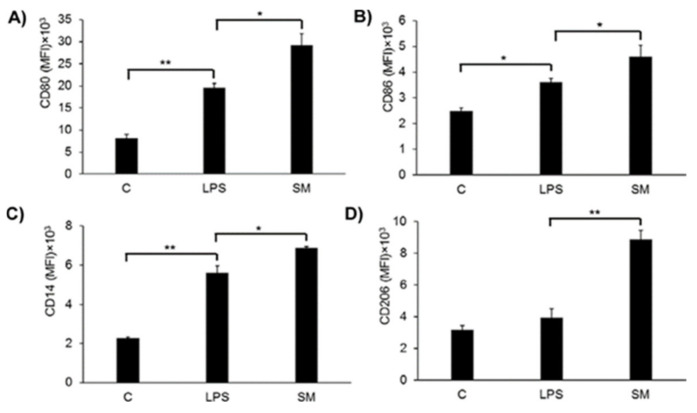
Effect of SM on the expression of cells surface markers CD80 (**A**), CD86 (**B**), CD14 (**C**), and CD 206 (**D**). Cells were incubated with LPS (1µg/mL) or SM (0.5 µM) for 24 h and stained with specific antibodies. The expression levels of specific surface molecules are presented in terms of mean fluorescent intensity (MFI). In the all experiments, non-treated cells was used as a control (designated by C). Values are expressed as the mean ± SD of the three independent experiments. * *p* < 0.05, ** *p* < 0.01.

**Figure 8 ijms-21-07876-f008:**
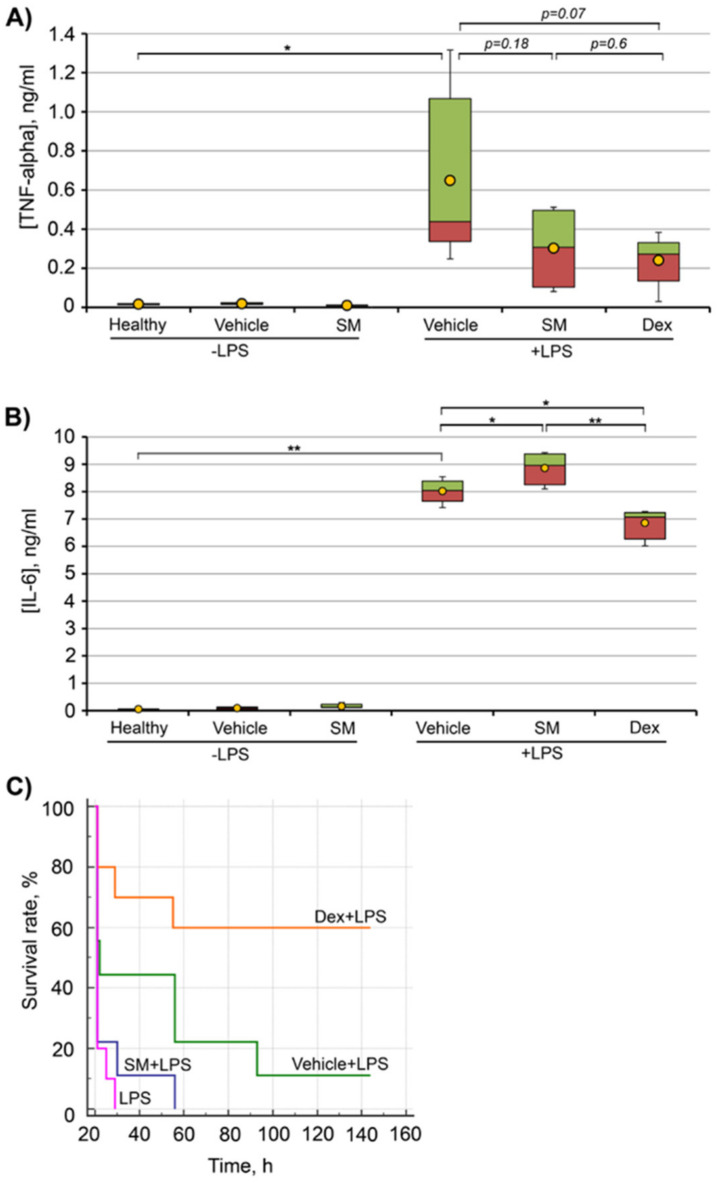
Anti-inflammatory effect of SM on the LPS-induced endotoxemia model. ICR mice were intraperitoneally (i.p.) administered with SM in 10% Tween-80 (vehicle) at a dose of 50 mg/kg 1 h prior to endotoxemia induction by LPS at a dose of 5 mg/kg (non-lethal endotoxemia) or 20 mg/kg (lethal endotoxemia). Pro-inflammatory cytokines TNF-α (**A**) and IL-6 (**B**) were measured by ELISA 4 h after non-lethal endotoxemia induction. Survival rates (**C**) were evaluated after lethal endotoxemia induction. In both experiments, dexamethasone at a dose of 0.5 mg/kg (**A**,**B**) and 1 mg/kg (**C**) was used as the reference drug. Differences between groups were statistically significant at * *p* < 0.05 and ** *p* < 0.005.

**Figure 9 ijms-21-07876-f009:**
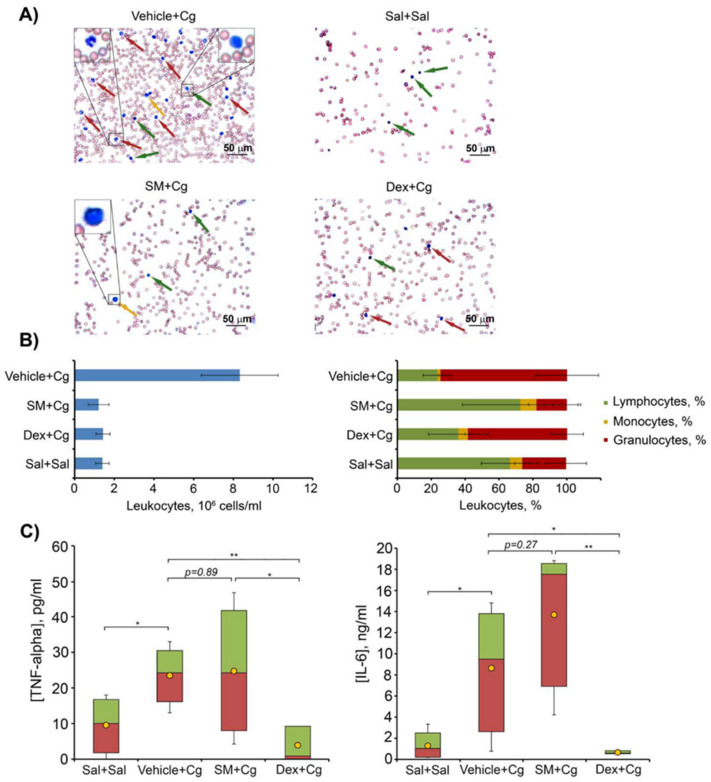
Anti-inflammatory effect of SM on the carrageenan-induced peritonitis model. ICR mice were intraperitoneally (i.p.) administered with SM in 10% Tween-80 (vehicle) at a dose of 50 mg/kg 1 h prior to peritonitis induction by 1% carrageenan followed by the counting of leukocytes in peritoneal fluid 4 h after carrageenan injection. Dexamethasone (0.5 mg/kg) was used as a reference drug. (**A**) The distribution of leukocyte subpopulations was evaluated by light microscopy after azur-eosin staining by the Romanovsky–Giemsa method. Red arrows show neutrophils, green arrows show lymphocytes, and yellow arrows show monocytes. Magnification ×400. (**B**) Total (left panel) and differential (right panel) leukocyte counts in peritoneal exudates. (**C**) Pro-inflammatory cytokines TNF-α and IL-6 were measured by ELISA 4 h after peritonitis induction. Differences between groups were statistically significant at * *p* < 0.05 and ** *p* < 0.005.

**Figure 10 ijms-21-07876-f010:**
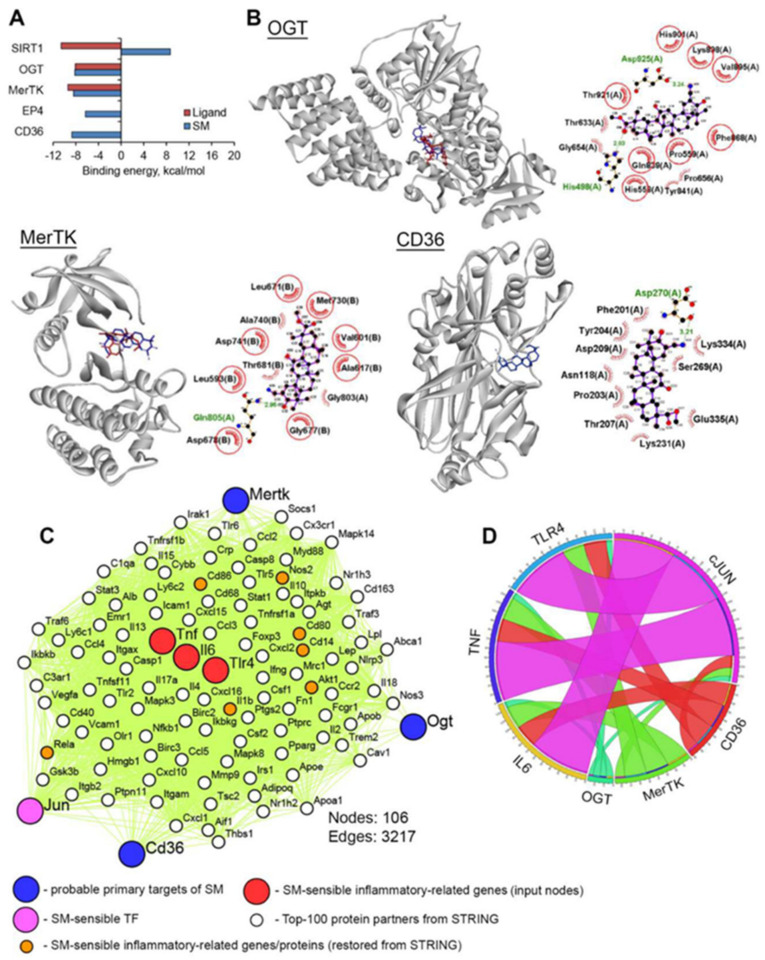
OGT, MerTK, and CD36 are probable primary targets of SM. (**A**) Binding energies of SM and proteins for which the modulation of activity/expression is associated with the LPS-induced inflammatory response in macrophages, according to data mining analysis. The binding energies were calculated using Autodock Vina. (**B**) The modes of binding of SM to OGT, MerTK, and CD36. Stereo presentation of the docked poses of SM in the listed proteins superimposed on the corresponding inhibitor-bound structures (OGT, MerTK) or phosphatidylserine recognition region (CD36) was created using BIOVIA Discovery Studio. Structures of SM and inhibitors are depicted as blue and red sticks, respectively. The 2D representations of docked poses of SM in mentioned proteins were reconstructed by LigPlot+. The green dashed lines and combs represent hydrogen bonds and non-direct interactions, respectively. Common amino acids, interacting with both SM and inhibitors, are highlighted in red circles. (**C**) The probable primary targets of SM and their association with SM-susceptible genes. The protein–protein interaction network was reconstructed using the STRING database (confidence score > 0.7, number of interactors = 100) and visualized by Cytoscape. (**D)** The number of the shortest paths between revealed probable protein targets of SM and pro-inflammatory genes in the protein–protein interaction (PPI) network. Only paths with length ≤ 1 node were calculated using the Pesca plugin.

**Table 1 ijms-21-07876-t001:** Predicted probable targets * of SM.

ID	Name	Relation to LPS-Induced Inflammation	Ref.
SIRT1	Sirtuin 1	Sirtinol (SIRT1 inhibitor) or knocking down SIRT1 by SIRT1 shRNA augmented LPS-induced TNF-α release by immortalized rat Kupffer RKC1 cells	[141]
		Down-regulation of SIRT1 expression through inhibition of SIRT1 activity using Ex527 and sirtinol enhanced LPS-induced TLR4 expression in rat renal inner medullary collecting duct cells	[142]
OGT	O-Linked N-acetylglucosamine (GlcNAc) transferase	OSMI-1 (selective inhibitor of OGT) significantly enhanced LPS-induced expression of IL-6 and TNF-α in murine bone marrow-derived macrophages	[143]
MerTK	MER receptor tyrosine kinase	MerTK-specific blocking antibody promoted LPS-induced production of TNF-α, IL-6, and IL1β in RAW264.7 cells	[144]
		Macrophages isolated from MerTK knockdown (MerTKKD) mice have been shown to be hypersensitive to bacterial LPS	[145]
EP4	Prostaglandin E receptor 4	EP2 agonist ONO-AE1-259 enhanced LPS-induced release of IL-6 by primary rat liver macrophages	[146]
CD36	Thrombospondin receptor	Direct binding of ursolic acid to CD36 induced release of IL-1β from murine peritoneal macrophages	[147]

* text mining analysis was performed by using the Google Scholar search engine based on the equation: [“enhanced LPS” OR “promoted LPS” OR “aggravated LPS” OR “exacerbated LPS”]. The articles published in the last 20 years were analyzed.

**Table 2 ijms-21-07876-t002:** The parameters of the molecular docking simulations.

Protein	PDB ID	Centre			Size		
		x	y	z	x	y	z
SIRT1	4I5I	43.323	−20.576	18.462	20	20	20
OGT	6MA3	−1.55	−43.134	16.031	22	20	20
MerTK	5U6C	−4.614	17.118	−18.308	20	20	20
EP4	5YWY	−42.732	−44.411	0.0	22	18	20
CD36	5LGD	−43.253	−26.463	25.028	50	50	70

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
