# Peer review of "Dual Effect of Soloxolone Methyl on LPS-Induced Inflammation In Vitro and In Vivo"

_ijms, 2020, doi:10.3390/ijms21217876_

Round 1

Reviewer 1 Report

The research group presents interesting data aimed to investigate the anti-inflammatory and anti-oxidative activities of a semisynthetic derivative of 18βH-GA, soloxolone methyl (methyl 2-cyano-3,12-dioxo-18βH-olean-9(11),1(2)-dien-30-oate, or SM) in vitro on LPS-stimulated RAW264.7 macrophages and in vivo in models of acute inflammation. The paper highlights an important topic and present novel finding in which SM significantly enhance LPS-induced expression of the pro-inflammatory cytokines IL-6, TNF-α 28 and IL-1β in RAW264.7 cells via activation of the c-Jun/TLR4 signalling axis. The manuscript is well designed with appropriate methodology and statistical analysis. Quality of data is adequate and consistent with the conclusions. 

Author Response

Thank you for your revision

Reviewer 2 Report

In this study, Markov et. al. found SM is an anti-inflammatory agent which have the ability to inhibit LPS-induced inflammation in vitro and in vivo. Even though the general idea is interesting, the authors need to prove the following:

  1. Cell viability is mainly caused by anti-proliferation or cytotoxicity (cell necrosis). What is the major mechanism of the effect of SM on the viability (Figure 1)?
  2. SM is able to induce HO-1and GSH expression (Figure 2). What is mechanism of SM to induced HO-1and GSH expression?
  3. Same as above, SM is able to inhibit phagocytosis but not cell migration without LPS treatment. (Figure 3). Since AKT and IKBa signals were not affect by SM without LPS treatment. (Figure 6 B and C) Is there any different signaling pathway for these results?
  4. The conclusion in Figure 8 (OGT, MerTK and CD36 are probable primary targets of SM) is mainly rely on docking study. It will be better if authors really test these targets involved in SM signalling?
  5. What is the rationale for authors choose the 50mg/Kg SM used in animal studies? (Figure 9 and 10) Is there any conversion formula about it?
  6. In year 2018, authors found SM was able to suppress the development of edema in a mouse model of carrageenan- or histamine- induced acute inflammation (Novel Glycyrrhetinic Acid Derivative Soloxolone Methyl Inhibits the Inflammatory Response and Tumor Growth). In this MS, they use the carrageenan-induced peritonitis model to test the anti-inflammatory effect of SM. Since the mechanism of carrageenan induced inflammation response is to release of histamine/5-hydroxytryptamine (5-HT)/bradykinin first and the second phase was to elevated production of prostaglandins or inducible cyclooxygenase (COX-2). These effects were more complicate than [another independent model of TLR4-mediated inflammatory response in vivo](Line 713-714). It will be better if authors discuss it more in discussion section.

Author Response

Point 1. Cell viability is mainly caused by anti-proliferation or cytotoxicity (cell necrosis). What is the major mechanism of the effect of SM on the viability (Figure 1)?

Response 1. Earlier [Logashenko, E.B. et al., Synthesis and Pro-Apoptotic Activity of Novel Glycyrrhetinic Acid Derivatives. ChemBioChem 2011, 12, 784–794] we showed that in cancer cells SM induces cell–cycle arrest, the translocation of phosphatidylserine to the cell surface, nucleus fragmentation, mitochondrial transmembrane potential dissipation and caspase activation. So the obtained data indicates, that SM induces death of cancer cells by caspase–dependent intrinsic pathway of apoptosis. We suppose that this mechanism of cell death underlies in the case of RAW264.7 cell line. 

Point 2. SM is able to induce HO-1and GSH expression (Figure 2). What is mechanism of SM to induced HO-1and GSH expression?

Response 2. Previously it was shown that cyano-enone-bearing triterpenoids are powerful inducers of the transcription factor Nrf2 and Nrf2-dependent activation of the endogenous antioxidant machinery [118–120,124,125,130]. It was shown that the anti-inflammatory and antioxidative actions of an entire set of synthetic triterpenoids, including CDDO, are closely linked and that the Nrf2/ARE system seems to provide a common mechanism for these activities of the triterpenoids. [Dinkova-Kostova, A.T. et al. Extremely potent triterpenoid inducers of the phase 2 response: correlations of protection against oxidant and inflammatory stress. Proc Natl Acad Sci U S A 2005; 102: 4584–9]. So, we suppose that the observed up-regulation of HO-1 and increased production of GSH in SM-treated  RAW264.7 cells (Fig. 3A,B) can be explained by the activation of Nrf2 pathway (please, see line 490-493, marked by yellow).

Point 3. Same as above, SM is able to inhibit phagocytosis but not cell migration without LPS treatment. (Figure 3). Since AKT and IKBa signals were not affect by SM without LPS treatment. (Figure 6 B and C) Is there any different signaling pathway for these results?

Response 3. Indeed, treatment of macrophages with SM without LPS challenge inhibited phagocytosis (Fig. 4A) and cellular motility (SM 0.5 µM, 19-46 h (Fig. 4B)), however, did not affect NF-κB and Akt signaling pathways in RAW264.7 cells. We suppose that observed inhibitory effects of SM on mentioned activities were NF-κB- and Akt-independent and can be explained by its interaction with actin-related protein 3 (Arp3) with subsequent dysregulation of microtubule organization, playing a key role in both phagocytosis and migration activity. Previously, To et al. showed that SM’s analog CDDO-Im at similar concentration (1 µM) binds to Arp3 and disorganized branched actin polymerization [1]. We described this suggestion in Discussion section (please, see lines 501-510). In order to make this statement more clear, some corrections were introduced into the mentioned paragraph (please, see lines 506-514, marked by yellow).

Point 4. The conclusion in Figure 8 (OGT, MerTK and CD36 are probable primary targets of SM) is mainly rely on docking study. It will be better if authors really test these targets involved in SM signalling?

Response 4. Indeed, patterns of SM interactions with OGT, MerTK and CD36 and their hypothetical master regulatory role in SM’s immunomodulatory activity were obtained by in silico methods, including text mining approach, molecular docking and network analysis. Due to the fact that immunostimulatory mechanism of cyano-enone-bearing triterpenoids (CETs) is virtually unknown (please, see Discussion), the main task of our in silico study was to identify probable directions of further investigations of this problem for any research groups dealing with CETs and other semisynthetic triterpenoids. Certainly, in a nearest future we are going to validate these data by biochemical and/or biophysical methods. In order to show more clearly that revealed data are based only on computer modeling, some corrections were introduced into the final paragraph of the chapter 2.7 (please, see lines 332-334, marked by yellow).

Point 5. What is the rationale for authors choose the 50mg/Kg SM used in animal studies? (Figure 9 and 10) Is there any conversion formula about it?

Response 5. In present work we used SM at a dose of 50 mg/kg since in our previous work for evaluation of anti-inflammatory activity of SM on the models of carrageenan- and histamine-induced paw edema we found out that dose of 50 mg/kg is the maximum possible dose for SM that can be administered in mice without significant toxicity (please, see lines 828-831, marked by yellow).

Point 6. In year 2018, authors found SM was able to suppress the development of edema in a mouse model of carrageenan- or histamine- induced acute inflammation (Novel Glycyrrhetinic Acid Derivative Soloxolone Methyl Inhibits the Inflammatory Response and Tumor Growth). In this MS, they use the carrageenan-induced peritonitis model to test the anti-inflammatory effect of SM. Since the mechanism of carrageenan induced inflammation response is to release of histamine/5-hydroxytryptamine (5-HT)/bradykinin first and the second phase was to elevated production of prostaglandins or inducible cyclooxygenase (COX-2). These effects were more complicate than [another independent model of TLR4-mediated inflammatory response in vivo](Line 713-714). It will be better if authors discuss it more in discussion section.

Response 6. In our study, we used two in vivo models of inflammation with TLR4-dependent mechanism, which allow us to assess the effect of SM on the different phases of inflammatory response, which involves a sequential changes both in the tissues and the whole organism, such as vasodilation, increased vascular permeability, recruitment of leukocytes, local and general increase in the level of cytokines and chemokines. As for carrageenan-induced peritonitis, the inflammatory response to carrageenan consisted of the several successive phases (Morris C.J., 2003; Phumsuay R. et al, 2020). The primary phase mediated by histamine/5-hydroxytryptamine is followed by a secondary kinin-mediated phase. The final phase is attributed to local production of prostaglandins (PGs) derived from arachodonic acid by the action of cyclooxygenase (COX) enzymes. A role for neutrophil derived reactive oxygen species, nitric oxide, and peroxynitrite in carrageenan-induced inflammation has also been identified. This information is included in the discussion section (please, see lines 645-649, marked by yellow).

Reviewer 3 Report

This manuscript constitutes a very interesting and detailed analysis of the anti-inflammatory activities of a semi-synthetic derivative of 18ßH-glycyrrhetinic acid and conclude to the high potential of this molecule in the treatment of local inflammation.

The study is very well conducted and the conclusions of each in vitro and in vivo test are relevant.

To my point of view, this manuscript is very easy and interesting to read.  It should be published with a few minor revisions.

  • for a better understanding of the text, I suggest to add in figure 1 the structures of oleanolic acid and 18ßH-GA
  • the references are not homogeneous (year of publication in bold)

Overall, very nice manuscript

Author Response

Point 1. For a better understanding of the text, I suggest to add in figure 1 the structures of oleanolic acid and 18ßH-GA

Response 1. Corrected. We added in figure 1 the structure of CDDO and 18ßH-GA

Point 2. The references are not homogeneous (year of publication in bold)

Response 2. Corrected

Round 2

Reviewer 2 Report

I appreciate that authors add changes according to the indication I provided. Nevertheless, some data are still not able to demonstrate what they claim. In order to demonstrate the interactions with OGT, MerTK and CD36, they should conduct some experiment to double check the effect but not only add introduction into the final paragraph of the chapter 2.7. Therefore, it is hard to accept this revision without further confirm the targets really involved in SM signalling. Of course, is up to the Editor to finally accept or not. 

Author Response

Dear Reviewer #2,

We sincerely thank you for the analysis of our revised manuscript and share your concern that in silico data should be double checked. Actually, our reconstruction of protein-protein interaction (PPI) network is aimed to independently verify the probable linkage between our molecular modeling data and revealed synergistic effects of SM with LPS on inflammatory response in macrophages. The obtained regulome, based on STRING database (Fig. 10), includes not only the input list of proteins (including probable targets of SM, c-Jun/TLR4 and IL-6/TNFα), but also other SM-sensitive genes/proteins, like p65 subunit of NF-κB, Akt, CD14, etc. Considering obtained independent results (molecular docking and network analysis), we speculated that proposed molecular mechanism of immunostimulatory activity of SM can be realized in the macrophages. It should be noted that the main and only aim of our computer modeling was to explain the molecular mechanisms of the novel activity of cyano-enone-bearing triterpenoids, which virtually has not yet been described previously, and to generate new ideas for further work in this field.

Indeed, we plan to estimate the direct binding of SM with OGT, MerTK and CD36 in a future experiments, in particular, by using surface plasmon resonance and\or isothermal titration calorimetry. However, it will take significant time and efforts and we plan to publish it separately when ready. We suppose that our existing data as it is now will be helpful for other research groups investigated the bioactivities of cyano-enone-bearing triterpenoids to understand probable unexpected results, especially obtained in animal studies and clinical trials (for instance, for group of Prof. Michael Sporn (Dartmouth Medical School, USA), which, according to NIH ProjectReporter (https://projectreporter.nih.gov/), are going to investigate analogs of CDDO for chemoprevention of COVID-19, the severity of which is tightly associated with cytokine storm induction (Project No. 3R43CA243842-01S1, start at 9-SEP-2020), for group of Prof. Karen Liby (Michigan State University, USA), which plans to study the effects of Nrf2 activators (including cyano-enone-bearing triterpenoids) on immune cells (Project No. 5R01CA226690-02, start at 30-MAR-2020), for group of Prof. Yongkui Jing (Shenyang Pharmaceutical University, China), which evaluates the molecular mechanisms of antitumor activity of cyano-enone-bearing triterpenoids, or group of Prof. Oxana Kazakova (Ufa Institute of Chemistry UFRC RAS, Russia), which develops novel selective enzyme inhibitors based on triterpenoid scaffold). Thus, in our opinion, the obtained in silico data should be presented in the current manuscript. However, considering your comment/recommendation, we translocated the description of these data from Results to Discussion section and modified it (please, see lines 524-580, yellow color). Thus, we discuss our molecular modeling data only as probable explanations of observed results, to bring new ideas for further studies. Moreover, in order to avoid misleading the potential readers of our work, the phrase that probable direct interactions of SM with OGT, MerTK and CD36 can underlie its observed immunostimulatory effects in macrophages was deleted from both Abstract and Conclusion sections (please, see lines 26 and 813, colored by yellow).

We hope that this version of the manuscript will be acceptable for publication.

Thank you!

Round 3

Reviewer 2 Report

No further questions